# Infectious stimuli promote malignant B-cell acute lymphoblastic leukemia in the absence of AID

Guillermo Rodríguez-Hernández [1,2,14], Friederike V. Opitz[3,14], Pilar Delgado [4,14], Carolin Walter[5], Ángel F. Álvarez-Prado [4], Inés González-Herrero[1,2], Franziska Auer[3,6], Ute Fischer[3], Stefan Janssen [3], Christoph Bartenhagen[7,8], Javier Raboso-Gallego[1,2], Ana Casado-García[1,2], Alberto Orfao[2,9], Oscar Blanco[2,10], Diego Alonso-López [11], Javier De Las Rivas [2,12], Sara González de Tena-Dávila[1,2], Markus Müschen [6], Martin Dugas [5], Francisco Javier García Criado[2,13], María Begoña García Cenador[2,13], Carolina Vicente-Dueñas [2], Julia Hauer[3,15]*, Almudena R. Ramiro[4], Isidro Sanchez-Garcia [1,2,15]* & Arndt Borkhardt [3,15]*

The prerequisite to prevent childhood B-cell acute lymphoblastic leukemia (B-ALL) is to decipher its etiology. The current model suggests that infection triggers B-ALL development through induction of activation-induced cytidine deaminase (AID; also known as AICDA) in precursor B-cells. This evidence has been largely acquired through the use of *ex vivo* functional studies. However, whether this mechanism governs native non-transplant B-ALL development is unknown. Here we show that, surprisingly, AID genetic deletion does not affect B-ALL development in Pax5-haploinsufficient mice prone to B-ALL upon natural infection exposure. We next test the effect of premature AID expression from earliest pro-B-cell stages in B-cell transformation. The generation of AID off-target mutagenic activity in precursor B-cells does not promote B-ALL. Likewise, known drivers of human B-ALL are not preferentially targeted by AID. Overall these results suggest that infections promote B-ALL through AID-independent mechanisms, providing evidence for a new model of childhood B-ALL development.

[1] Experimental Therapeutics and Translational Oncology Program, Instituto de Biología Molecular y Celular del Cáncer, CSIC/Universidad de Salamanca, Salamanca, Spain. [2] Institute of Biomedical Research of Salamanca (IBSAL), Salamanca, Spain. [3] Pediatric Oncology, Hematology and Clinical Immunology, Medical Faculty, Heinrich Heine University, Moorenstrasse 5, 40225 Düsseldorf, Germany. [4] B Cell Biology Lab, Centro Nacional de Investigaciones Cardiovasculares, Madrid, Spain. [5] Institute of Medical Informatics, University of Muenster, Muenster, Germany. [6] Department of Systems Biology, City of Hope Comprehensive Cancer Center, Monrovia, California 91016, USA. [7] Department of Experimental Pediatric Oncology, University Children's Hospital of Cologne, Medical Faculty, Cologne, Germany. [8] Center for Molecular Medicine Cologne (CMMC), University of Cologne, Cologne, Germany. [9] Servicio de Citometría, Departamento de Medicina, CIBERON, and Instituto de Biología Molecular y Celular del Cáncer, CSIC/Universidad de Salamanca, Salamanca, Spain. [10] Departamento de Anatomía Patológica, Universidad de Salamanca, Salamanca, Spain. [11] Bioinformatics Unit, Cancer Research Center (CSIC-USAL), Salamanca, Spain. [12] Bioinformatics and Functional Genomics Research Group, Cancer Research Center (CSIC-USAL), Salamanca, Spain. [13] Departamento de Cirugía, Universidad de Salamanca, Salamanca, Spain. [14] These authors contributed equally: Guillermo Rodríguez-Hernández, Friederike V. Opitz, Pilar Delgado. [15] These authors jointly supervised this work: Julia Hauer, Isidro Sanchez-Garcia, Arndt Borkhardt. *email: Julia.Hauer@med.uni-duesseldorf.de; isg@usal.es; Arndt.Borkhardt@med.uni-duesseldorf.de

The most common cancer in childhood is B-cell precursor acute lymphoblastic leukemia (B-ALL)[1,2]. Although the overall survival is excellent, therapeutic options can be associated with severe toxic side effects, and 20% of children still relapse[3]. Thus, prevention approaches are undoubtedly preferred to any therapy advance. The prerequisite to build strategies for preventing B-ALL is to discover its etiology[4]. Somatic lesions of the B-cell transcription factor gene *PAX5* are characteristic of B-ALL[5,6], and this key role of PAX5 in the genesis of B-ALL has been even broaden by recent discoveries of inherited *PAX5* mutations associated to a syndrome of susceptibility to B-ALL[7,8]. The presence of the *PAX5* genetic alteration seems to create a hidden preleukemic clone that remains latent until it is later triggered by environmental stimuli[9]. Chronic infections during early childhood were previously implicated in the etiology of childhood B-ALL[10–12]. We have recently showed that natural exposure to infectious pathogen induced development of overt B-ALL in mice mimicking human preleukemic lesions, like Pax5-haploinsuffiency or *ETV6-RUNX1* fusion gene[4,13].

Activation-induced deaminase (AID) plays a central role in the immune response by triggering somatic hypermutation (SHM) and class-switch recombination in germinal center B cells[14]. In addition, AID is required for germinal center-derived lymphomagenesis[15–19] and a recent mouse model of endemic Burkitt lymphoma, which is caused by chronic infection, identified AID triggered infection-driven B-cell lymphomagenesis[20]. AID is not generally expressed in early bone marrow B-cell precursors[21]. However, the current view in the field of B-cell leukemogenesis states that AID expression is induced in preleukemic B-cell precursor cells in response to infection and promotes in this case secondary genetic changes that may lead to subsequent leukemia development. However, evidence supporting this model has been largely acquired through the use of ex vivo functional studies involving bone marrow transplantation[22–25]. Whether AID also contributes to native (non-transplant) B-ALL development is to date entirely unclear. Based on these observations, we examined here whether AID is required for clonal evolution of pre-malignant precursor B cells in the etiology of B-ALL by using both loss-of-function and gain-of-function genetic approaches. Overall, our results suggest that infectious stimuli can promote malignant B-cell leukemogenesis through AID-independent mechanisms.

## Results

**In vivo Aid expression in preleukemic precursor B cells**. AID is responsible for the induction of secondary diversification of immunoglobulin (Ig) genes in secondary lymphoid organs in response to antigen. AID initiates SHM and also Ig class switching, but it can also promote chromosomal translocations and mutations with an etiological role in B-cell lymphomagenesis[16–19,26]. We have recently shown that exposure to natural infectious pathogen triggered clonal evolution toward B-ALL[4,13]. Based on these findings, we asked whether AID is required for clonal evolution of pre-malignant precursor B cells in the etiology of native (non-transplant) infection-associated B-ALL. Thus, we first investigated if high levels of *AID* were present in in vivo preleukemic precursor B cells purified from mice carrying a genetic susceptibility to B-ALL (either *Pax5* heterozygosity or the presence of the *ETV6-RUNX1* fusion gene), which are exposed to natural infections (Supplementary Fig. 1a). Both mouse models only develop B-ALL under natural infection exposure[4,13]. In agreement with previous results[21], AID was not detectable in preleukemic precursor B cells isolated from the bone marrow (BM) of mice kept under SPF (germ-free) conditions (Supplementary Fig. 1a). Similarly, *AID* expression levels were not upregulated in preleukemic precursor B cells isolated from BM of mice kept in conventional (natural infection) housing (Supplementary Fig. 1a). In a previous study[22], withdrawal of IL7 and repeated ex vivo exposure of Fraction D pre-B cells to inflammatory stress (LPS) resulted in high levels of *AID* mRNA and protein expression. However, the exposure to natural infection does not significantly increase *AID* expression in preleukemic precursor B cells (Supplementary Fig. 1a), although in vitro exposure of preleukemic *Pax5-het*, and *Sca1-ETV6-RUNX1* precursor pro-B cells to different immune activation stimuli resulted in high levels of AID mRNA (Supplementary Fig. 1b).

**Natural infections drive B-ALL in the absence of AID**. Given the clonal nature of leukemia, we cannot exclude that Aid would be overexpressed at a single preleukemic precursor B cell in our model for in vivo exposure to natural infection. To test a causative role of AID in native infection-driven B-ALL development, we employed *Aicda/Aid-KO* mice to study loss of AID function in the clonal evolution of pre-malignant B cells in the etiology of infection-associated B-ALL. *PAX5* is a gene commonly mutated in B-ALL[6]. *Pax5-het* mice are prone to develop B-ALL only after natural infection exposure[4]. Likewise, *Pax5-het* mice used as sentinels of an ongoing similar infectious stimuli within the facilities are permanently kept either in a specific pathogen-free (SPF) animal house during their lifespan or transferred to natural infection exposure as described[4], where the microbiologic status is defined and controlled (Supplementary Tables 1, 2). *Pax5-het* mice housed under SPF environment have not spontaneously developed B-ALL unless were exposed to natural infection environment, corroborating previous results[4]. Thus, *Pax5-het/Aid-het* and *Pax5-het/Aid-KO* animals were exposed to identical natural infection environment (Fig. 1a; Supplementary Tables 1, 2), and B-ALL development was monitored and analyzed in these strains as previously described[4]. Surprisingly, under this scenario, B-ALL development was observed in 30.4% (7 out of 23) of *Pax5-het/Aid-het* and in 30% (9 out of 30) of *Pax5-het/Aid-KO* animals, respectively (Fig. 1b), closely resembling the penetrance of B-ALL development in both the *Pax5-het* group[4] and the carriers with the heterozygous *PAX5* c.547G>A mutation[7]. The advent of leukemia in either *Pax5-het/Aid-het* or *Pax5-het/Aid-KO* background under natural infection exposure occurred at between 7 and 20 months of age (Fig. 1b), similar to the *Pax5-het* group[4]. These *Aid-het* or *Aid-KO* B-ALLs became manifest due to blast infiltration and presence of tumor cells in the peripheral blood (PB) (Fig. 1c; Supplementary Fig. 2). FACS analysis revealed the expected $CD19^{+/-}B220^{+}IgM^{-}cKit^{+/-}CD25^{+/-}$ cell surface phenotype for B-cell blasts that infiltrated both hematopoietic tissues like BM, PB, spleen, and lymph nodes (Fig. 1c; Supplementary Figs. 2–4) and nonlymphoid tissues, such as the liver and lung (Supplementary Figs. 3, 4). Likewise, both groups of B-ALL, *Pax5-het/Aid-het* and *Pax5-het/Aid-KO*, also presented a similar percentage of blast cells infiltrating BM, spleen, and PB (Supplementary Fig. 5). Similar to the *Pax5-het* group, all *Pax5-het/Aid-het* or *Pax5-het/Aid-KO* B-ALLs displayed clonal immunoglobulin $V_{H}-DJ_{H}$ and $D-J_{H}$ gene rearrangements consistent with a pro-B or pre-B-cell origin (Supplementary Fig. 6), and the DJ/VDJ junctions did not show specific features regarding the junctions or choice of V, D, and J segments (Supplementary Fig. 7). Likewise, we next identified differentially expressed genes (FDR = 0.01) in leukemic BM from both *Pax5-het/Aid-het* (Supplementary Fig. 8 and Supplementary Data 1) and *Pax5-het/Aid-KO* (Supplementary Fig. 9 and Supplementary Data 2) mice compared with BM precursor B cells from wild-type mice, showing that AID-deficient B-ALL exhibited a gene expression pattern that is similar to AID-het B-ALL (Supplementary Fig. 10).

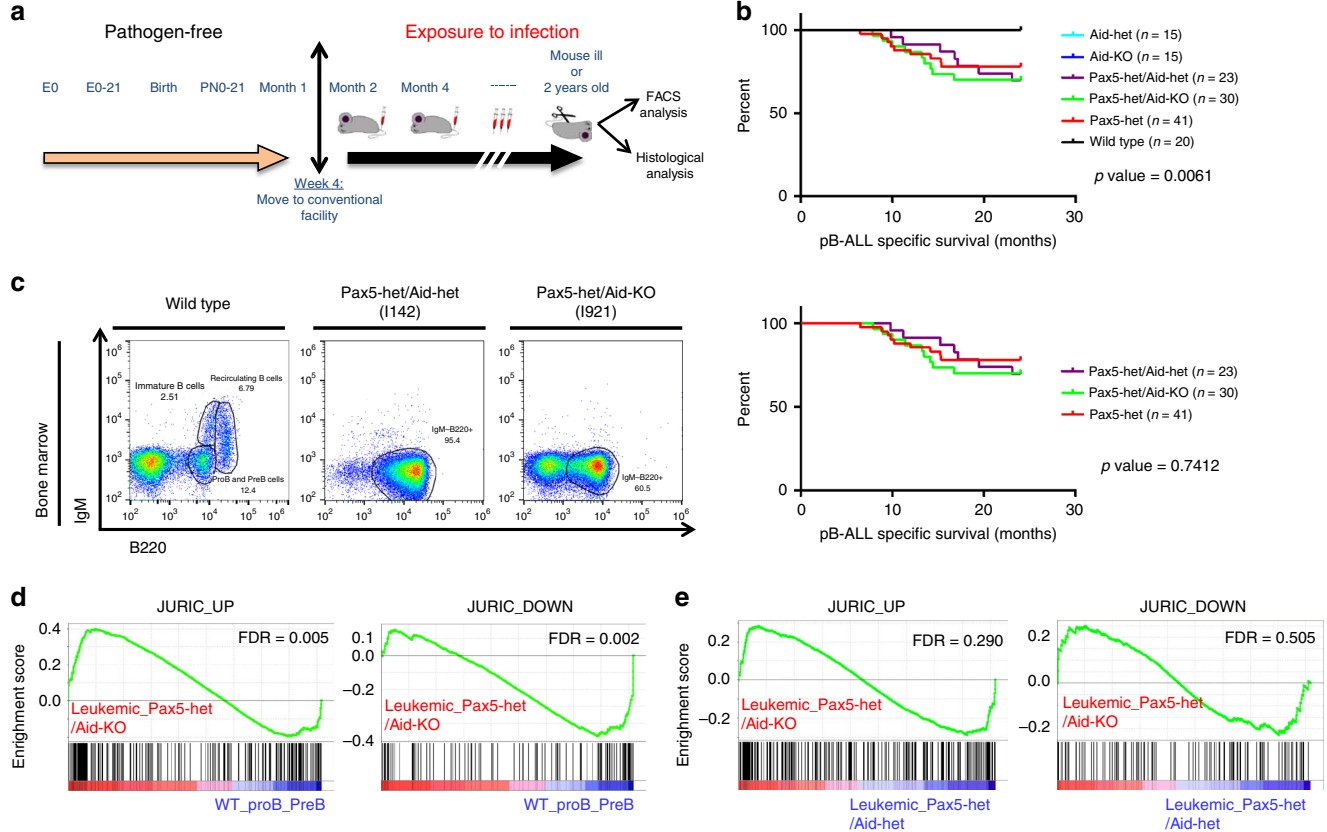

**Fig. 1 Contribution of AID to the genesis of native infection-driven B-ALL. a** Experimental design. Mice permanently kept either in a SPF animal house during their lifespan or transferred to natural infection exposure between 4 and 5 weeks after birth. **b** Comparison between B-ALL-specific survival of *Aid-het* mice (light blue, *n* = 15), *Aid-KO* mice (deep blue, *n* = 15), *Pax5-het/Aid-het* mice (violet, *n* = 23), *Pax5-het/Aid-KO* mice (green, *n* = 30), *Pax5-het* animals (red, *n* = 41), and WT control mice (black, *n* = 20). *Pax5-het/Aid-het* or *Pax5-het/Aid-KO* mice showed a significantly shortened lifespan (log-rank *p*-value = 0.0061) (upper panel) that was similar to the *Pax5-het* group (lower panel) (log-rank *p*-value = 0.7412). **c** Flow cytometric analysis of hematopoietic subsets in diseased *Pax5-het/Aid-het* and *Pax5-het/Aid-KO* mice. Comparison between leukemic BM showing conglomeration of blast B cells in *Pax5-het/Aid-het* mice (*n* = 7; age, 9–20 months) and *Pax5-het/Aid-KO* mice (*n* = 9; age, 7–17 months) and normal BM showing normal B cells in control littermate age-matched WT mice (*n* = 4; age, 8–16 months). **d** GSEA of leukemic *Pax5-het/Aid-KO* B cells. GSEA identified significant enrichment in human BCR-ABL B-ALL genesets[28] in *Pax5-het/Aid-KO* tumor-bearing BMs compared with sorted purified pro/pre-B cells from BM of WT mice (GSEA FDR = 0.005, and FDR = 0.002). **e** GSEA results showing no significant (FDR > 0.25) enrichment of human BCR/ABL B-ALL genesets[25] between leukemic *Pax5-het/Aid-het* and leukemic Pax5-het/Aid-KO phenotypes.

To explore the relevance of our findings in human leukemia, we compared the role of AID activity on mRNA expression derived from tumor cells from patients with B-ALL expressing or not AID[27,28]. Although AID-deficient B-ALL exhibited a gene expression pattern similar to these human B-ALL[28] (FDR = 0.005; FDR = 0.002) (Fig. 1d), the gene expression profiles between *Pax5-het/Aid-KO* B-ALL and *Pax5-het/Aid-het* B-ALL did not show any differences between them when we compared with gene expression profile derived from tumor cells from patients with B-ALL expressing or not AID[27,28] (FDR = 0.290; FDR = 0.505) (Fig. 1e), indicating that the AID is not significantly contributing to the leukemic gene expression profile of infection-driven B-ALL.

In order to further identify somatically acquired 2nd hits leading to leukemia development in both *Pax5-het/Aid-het* and *Pax5-het/Aid-KO* B-ALLs, we performed whole-exome sequencing of five *Pax5-het/Aid-het* tumors, eight *Pax5-het/Aid-KO* tumors, and paired samples of germline tail DNA as a reference. We could identify between 8 and 58 somatic single-nucleotide variations (SNVs) in the tumor samples of *Pax5-het/Aid-het* mice and between 11 and 27 somatic SNVs in the tumor samples of *Pax5-het/Aid-KO* mice. According to the cancer gene consensus database[29], the somatic cancer SNVs ended in 0–4 SNV per

mouse (Fig. 2a). Furthermore, we analyzed these present SNVs and could show that *Pax5-het/Aid-het* and *Pax5-het/Aid-KO* mouse tumors carry mutations in the same genes as well as mutations that are specific for the genotypes (Fig. 2b; Table 1). Similar genes (*Pax5*, *Trp53*, and *Ptpn11*) were mutated in both *Pax5-het/Aid-het* and *Pax5-het/Aid-KO* leukemic cells. Notably, the drivers of mouse infection-associated B-ALL are not preferentially targeted by AID (Supplementary Table 3). Consistently, tumor *Pax5-het/Aid-het* and *Pax5-het/Aid-KO* pro-B cells grow independent of IL7 and can initiate leukemia in secondary transplant recipients (Supplementary Fig. 11).

Furthermore, we analyzed the tumor mutational pattern and the mutation spectrum indicated a general prevalence of transitions between pyrimidines in the sample set (Fig. 3a). COSMIC signatures 1, 3, and 12 had the highest overall relative contribution[30], but no signature was specific for the tumor subsets (Fig. 3b). A de novo signature analysis revealed no significant differences between tumor and control groups (Fig. 3c, d); both signature candidates reflected the overall mutation spectrum. For this analysis, nonnegative matrix factorization (NMF) was applied as dimension reduction technique. The similarity between original and reconstructed mutation spectrum was high (cosine similarity $S_C$: $S_C$ (original, COSMIC) = 0.94 for

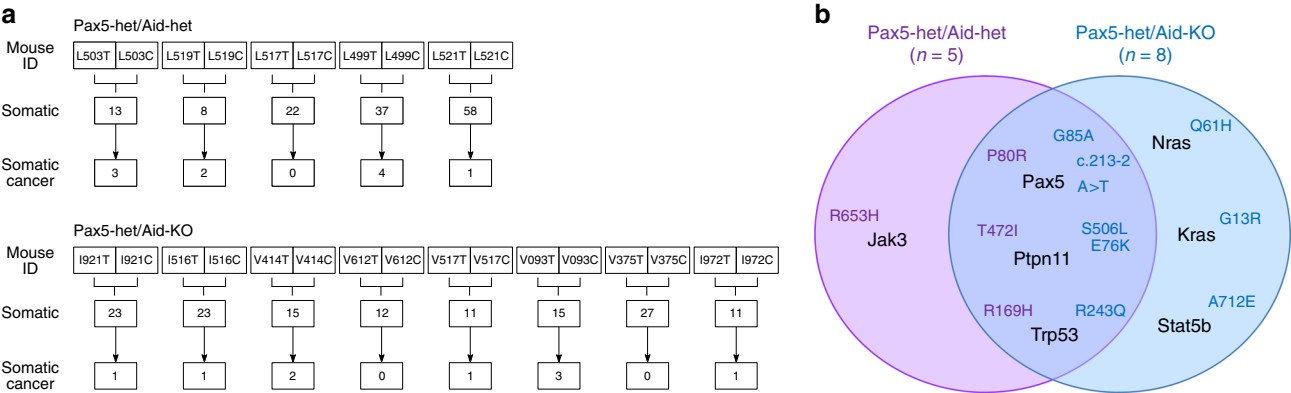

**Fig. 2 Mouse tumor exome sequencing in *Pax5-het/Aid-het* mice and *Pax5-het/Aid-KO* native B-ALL. a** Whole-exome sequencing analysis of tumor and control samples. Tumor-specific somatic mutations were determined by *mutect* and *varscan* analysis. The number of somatic cancer genes were calculated by using the cancer gene consensus list. **b** Genomic comparison between mutations driving native B-ALL as a result of natural infection exposure of *Pax5-het/Aid-het* (violet) and *Pax5-het/Aid-KO* mice (light blue), respectively, showed that similar second hits were affected by recurrent mutations.

the COSMIC decomposition, and $S_C$ (original, de novo) = 0.96 for the de novo analysis). These observations surprisingly revealed that clonal evolution of pre-malignant B-cell precursors toward B-ALL as a result of in vivo natural infection exposure took place through the acquisition of secondary genetic events unrelated to AID activity in mouse models mimicking human genetic predisposition to B-ALL.

**Premature Aid expression in B cells does not trigger B-ALL.** Subsequently, we investigated whether aberrant in vivo AID function generates driver mutations in BM precursor B cells with physiological RAG V(D)J recombinase. In order to achieve this, we made use of a conditional mouse model for *Aid* expression where a cassette harboring the *Aicda* and *EGFP* cDNAs separated by an internal ribosomal entry site (IRES) sequence was placed inside the Rosa26 endogenous locus. The *Aid-EGFP* cassette is preceded by a floxed transcriptional stop sequence that prevents *Aicda* expression (*Rosa26*$^{+/Aid}$ mice[31]; Supplementary Fig. 12). *Rosa26*$^{+/Aid}$ mice were bred to the Mb-1-Cre deleter strain[32] to promote AID expression from early pro-B-cell stages of B-cell differentiation. We found that aberrant AID expression in BM precursor B cells did not affect mouse survival or triggered B-cell neoplasia development (Fig. 4a), although we detected GFP expression starting at the pre-pro-B-cell stage and in virtually 100% of pro-B cells and at all subsequent differentiation stages (Fig. 4b). We next assessed AID activity in immunoglobulin (Ig) genes in splenic (Fig. 4c) and bone marrow B cells (Fig. 4d). We observed that pro-B cells seem to be the most permissive stage for AID activity since the SHM does not increase significantly later in B-cell differentiation (Fig. 4d).

In agreement with previous studies[33–36], we observed that premature expression of AID in precursor B cells did not induce B-cell malignancies. However, it was previously shown that the combination of AID transgenic expression driven by kappa light-chain regulatory sequences (IgκAID mice) with p53 deficiency increased the proportion of B-cell lymphomas compared with p53$^{−/−}$ mice[37]. These results were interpreted as an indication that AID-induced genetic lesions are under tight control of *Tp53* tumor suppressor that eliminates B cells that sustained extensive DNA damage. For this reason, we additionally bred *Rosa26*$^{+/Aid}$ mice to tumor p53 suppressor-deficient background (*Tp53*$^{−/−}$). However, Aicda overexpression did not affect the development of B-cell lymphomas (Fig. 4a). At present, we cannot tell whether this discrepancy is due to the higher AID expression levels achieved by the IgκAID transgene compared with our *Rosa26*$^{+/Aid}$ *Mb1-Cre*$^{ki/+}$ mice, or to the fact that IgκAID is expressed later during B-cell differentiation and induced upon B-cell activation, or related to the random genomic insertion occurring when classical transgenic procedures are used[33]. Regardless of the reason, our data show that the aberrant AID expression in *Rosa26*$^{+/Aid}$ *Mb1-Cre*$^{ki/+}$ mice that promotes intense off-target mutagenic activity initiated at the earliest stages of B-cell development is not sufficient to drive precursor B-cell transformation. Notably, this finding was also substantiated in humans, since AID off-targets[26] are not significantly enriched in human B-ALL drivers[38] (Supplementary Fig. 13 and Supplementary Table 4).

**Discussion**
The current view in the field of B-cell leukemogenesis states that AID expression is induced in preleukemic B-cell precursor cells in response to infection and promotes in this case secondary genetic changes that may lead to subsequent leukemia development[22–25]. The main experimental approach used to decipher the process of infection-driven leukemogenesis has relied on ex vivo functional assays involving transplantation, and it has been focused in a specific stage of B-cell development, the late pre-B cells at the time of κ-light-chain rearrangement[22]. Repeated in vitro forced stimulation with IL7-withdrawal to drive κ-light-chain gene rearrangement and LPS inflammatory stimuli of these late pre-B cells at the time of κ-light-chain expression resulted in a > 100-fold increase of AID protein levels[22]. Downregulation of IL7R signaling and concomitant activation of RAG1 were both required to promote malignant transformation[22]. In agreement with this observation, the combination of deregulated AID expression close to the loss of 53BP1 in late pre-B cells at the time of κ-light-chain expression gave rise to B-cell transformation[39]. However, it is not known to what extent this mechanism is shared by the pro-B or pre-B cells, stage of B-cell development which drives common B-cell precursor ALL, in the context of native (non-transplant) infection-driven B-ALL development[4,13,24]. In agreement with the previous observation[39], we found that in vitro exposure of preleukemic pro-B cells to different immune activation stimuli resulted in a significant increase of AID mRNA (Supplementary Fig. 1b). However, natural infection exposure does not increase Aid expression in preleukemic precursor B cells in vivo (Supplementary Fig. 1a), although these results cannot exclude that Aid might be overexpressed at single preleukemic precursor B cell as a result of in vivo exposure to natural infection. In this paper, we addressed this question using two novel genetic in vivo models: an experimental mouse model that develops B-ALL only in response to natural infection (the "*pax5-het*" model, which faithfully mimics human disease with respect

**Table 1 Genetic characteristics of *Pax5-het/Aid-het* (1–5) and *Pax5-het/Aid-KO* (6–13) B-ALL.**

| Mouse number | | Mouse age at disease (months) | Jak3 mutation in mouse* | JAK3 human homolog | Stat5b mutation in mouse* | Stat5b human homolog | Pax5 mutation in mouse* | Pax5 human homolog | Trp53 mutation in mouse* | Trp53 human homolog | Nras mutation in mouse* | Nras human homolog | Kras mutation in mouse* | Kras human homolog | Ptpn11 mutation in mouse* | Ptpn11 human homolog |
|---|---|---|---|---|---|---|---|---|---|---|---|---|---|---|---|---|
| 1 | L503 | 10 | R653H | R657Q | - | - | P80R | P80R | - | - | - | - | - | - | - | - |
| 2 | L519 | 15.4 | - | - | - | - | - | - | R169H | R175H | - | - | - | - | - | - |
| 3 | L517 | 17.4 | - | - | - | - | - | - | - | - | - | - | - | - | - | - |
| 4 | L499 | 17 | - | - | - | - | - | - | - | - | - | - | - | - | T472I | Not reported |
| 5 | L521 | 19.7 | - | - | - | - | - | - | - | - | - | - | - | - | - | - |
| 6 | I921 | 9.1 | - | - | - | - | G85A | G85R | - | - | - | - | - | - | - | - |
| 7 | I516 | 13.4 | - | - | - | - | - | - | - | - | - | - | - | - | S506L | S506L |
| 8 | V414 | 8 | - | - | - | - | - | - | R243Q | R249Q | - | - | - | - | E76K | E76K |
| 9 | V612 | 10.2 | - | - | - | - | - | - | - | - | - | - | - | - | - | - |
| 10 | V517 | 11.5 | - | - | - | - | - | - | - | - | Q61H | Q61K | G13R | G13R | - | - |
| 11 | V093 | 14.2 | - | - | A712E | V712E | c.213-2 A>T | c.213-1 G>T | - | - | - | - | - | - | - | - |
| 12 | V375 | 13.5 | - | - | - | - | - | - | - | - | - | - | - | - | - | - |
| 13 | I972 | 16.7 | - | - | - | - | - | - | - | - | - | - | - | - | - | - |

*Mutations confirmed by Sanger sequencing

to disease penetrance, clinical and genomic characteristics), and a complementary mouse model for conditional *Aid* expression in B-cell precursors. Using these two models, we clearly show that ex vivo approaches are not representative for the in vivo situation in this case. We demonstrate that Aid expression levels are not upregulated in pre-malignant pro-B or pre-B-cell precursors in mice prone to develop B-ALL under natural infection exposure. Similarly, enforced AID mutagenic activity in these pro-B and pre-B cells did not promote B-ALL development, and known drivers of human B-ALL are not preferentially targeted by AID. Likewise, patients with B-ALL have relatively few SNVs, and rarely have chromosomal translocations into the IGH switch region. In this regard, it is worth noting that AID activity is closely associated to transcription and usually occurs near transcription start sites and linked with multiple promoter and enhancer regions, most likely to facilitate AID accessibility[19,40,41]. Therefore, off-target AID activity is likely to differ between early B-cell precursors—driving leukemogenesis—and germinal center B cells—giving rise to mature B-cell lymphomas—according to the transcriptional configuration associated with each differentiation stage. Likewise, the presence or absence of AID does not impact on the latency or incidence of infection-mediated B-ALL development. This Aid-independently emerged B-ALL closely resembles childhood B-ALL in both clinical characteristics and leukemia-associated genetic alterations. Our study therefore provides evidence for a substantially revised model of leukemogenesis triggered by infection.

As opposed to AID-driven B-cell lymphomagenesis, where the penetrance of the disease is very high[20], the discovery that infection-mediated native B-ALL takes place without AID contribution is consistent with the fact that only a minority of healthy newborns harboring a preleukemic clone evolve to overt B-ALL[42]. This fact makes it more plausible that this infection-driven leukemogenesis process might be prevented[4,13,24,43]. In addition, these findings have also potential implications for the clinical use of PI3Kδ or Bruton's tyrosine kinase inhibitors in the treatment of B-cell leukemias[44,45]. The introduction of these inhibitors into the clinic could change how B-cell leukemias are treated[44]. However, the increase of AID levels induced by these inhibitors only after drugs treatment in vitro, and in tumor cell lines has been considered one of their major limitations as it may affect genomic stability in precursor B cells[46,47]. Furthermore, a significant AID activity on DNA of treated patients was only demonstrated for PI3Kδ. However, no B-ALL development has been reported so far by current use of targeted therapy with the PI3Kδ inhibitor in patients with activated PI3Kδ syndrome (APDS)[48]. In this regard, and in contrast with recent data derived from studies based on in vitro and in tumor cell lines[46], recent observations have demonstrated that ibrutinib therapy results in a profound reduction in both AID expression and proliferative leukemic fractions in treated patients[49]. Overall, this work highlights the unique potential of the application of this in vivo native assay to identify the importance of drivers of B-cell precursor ALL in the context of infectious exposure or any other environmental factors.

## Methods

**Mouse model for natural infection-driven leukemia.** *Pax5-het* mice were crossed with *Aid-KO* mice[14] to generate *Pax5-het/Aid-het* and *Pax5-het/Aid-KO*. These *Pax5-het/Aid-het* and *Pax5-het/Aid-KO* mice were bred and maintained in the SPF area of the animal house until the moment when they were relocated to an environment where natural infectious agents were present, as previously described[4]. All mouse experiments were performed following the applicable Spanish and European legal regulations, and had been previously authorized by the pertinent institutional committees of both University of Salamanca and Spanish Research Council (CSIC). *Aid-het*, *Aid-KO*, *Pax5-het/Aid-het*, and *Pax5-het/Aid-KO* mice of a mixed C57BL/6× CBA background were used in this study, with approximately

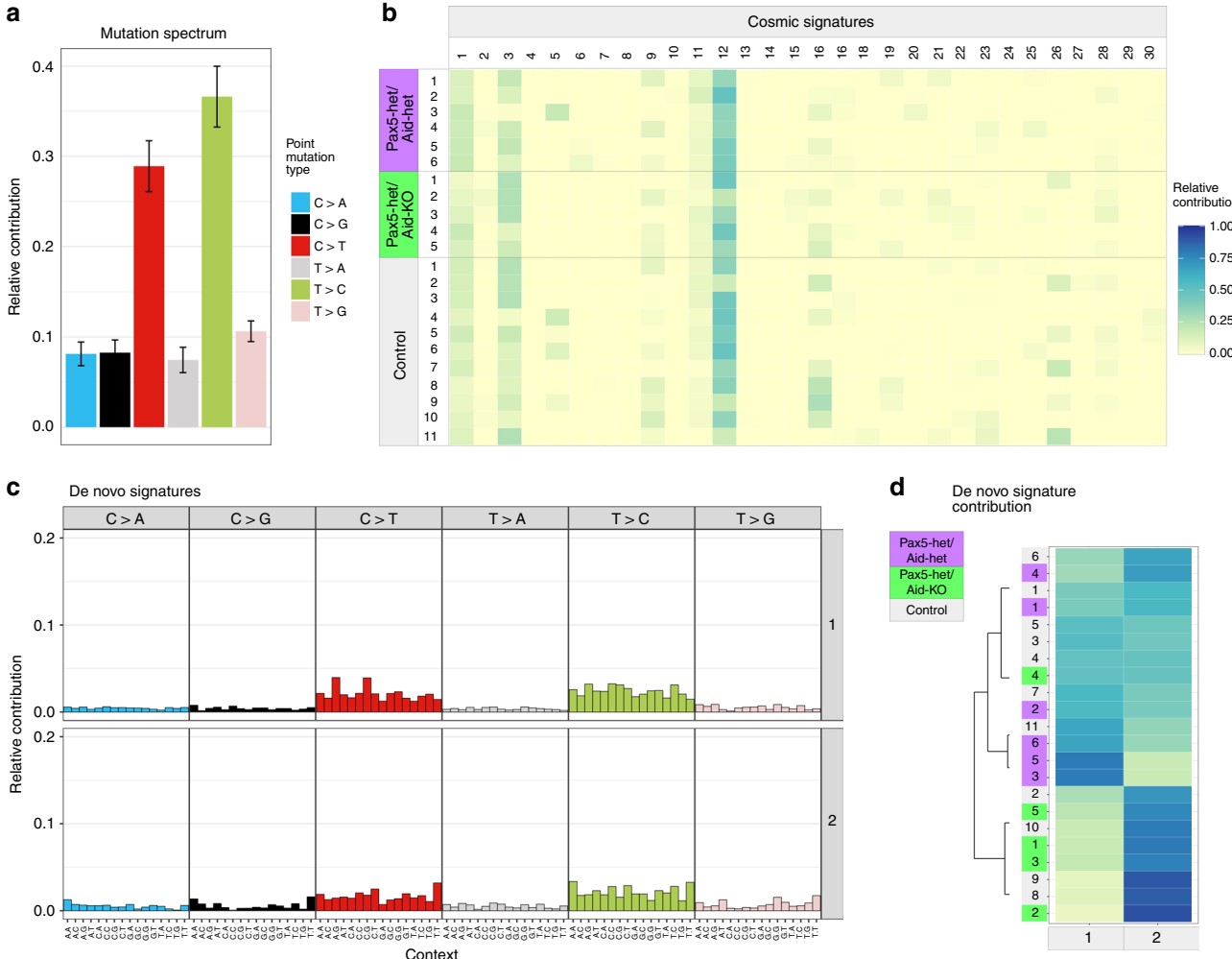

**Fig. 3 Mutational patterns in *Pax5-het/Aid-het* and *Pax5-het/Aid-KO* B-ALL. a** Mutation spectrum of total identified SNPs after filtering (median: 524 SNPs per sample). **b** Spectrum decomposition with COSMIC signatures as basis. COSMIC signatures 1, 3, and 12 showed the highest overall relative contribution, while no signature indicated tumor-specific enrichment. **c** De novo signature results for a nonnegative matrix factorization decomposition with two signatures. Enrichment of mutation contexts mirrored the original mutation spectrum for both signatures. **d** De novo signature contribution for signatures shown in (**c**), with clustering (complete-linkage). No tumor-specific cluster was identified.

equal representation of both males and females. For the experiments, *Aid-het, Aid-KO, Pax5-het/Aid-het*, and *Pax5-het/Aid-KO* of the same litter were used. When the animals showed evidences of illness, they were humanely killed, and the main organs were extracted by standard dissection. All major organs were macroscopically inspected under the stereo microscope, and then representative samples of tissue were cut from the freshly dissected organs, and were immediately fixed.

**Oncogenic priming of B cells by premature expression of Aid.** *Rosa26*[AID] knock-in mice were generated by our group and described previously[31]. *Rosa26*[AID] mice were crossed with *Mb-1*[cre][32], *P53*[−/−][37], and *Ink4/Arf*[−/−][50], respectively. *Aid-KO* mice[14] were used as negative control for SHM assays. Both males and females' littermates aged from 6 to 12 weeks were used for experiments. All animals were housed in the Centro Nacional de Investigaciones Cardiovasculares animal facility under SPF conditions. All animal procedures conformed to EU Directive 2010/63EU and Recommendation 2007/526/EC regarding the protection of animals used for experimental and other scientific purposes, enforced in Spanish law under RD 53/2013. The procedures have been reviewed by the Institutional Animal Care and Use Committee (IACUC) of Centro Nacional de Investigaciones Cardiovasculares, and approved by Consejeria de Medio Ambiente, Administración Local y Ordenación del Territorio of Comunidad de Madrid.

**Mutation analysis by next-generation sequencing (NGS).** Analysis of mutations at Sμ by NGS was performed as described in Pérez-Durán P et al.[51]. Briefly, naïve B cells were purified by immunomagnetic depletion with anti-CD43 beads (Miltenyi) and cultured with 25 μg/ml LPS (Sigma) plus 10 ng/ml IL4 (PeproTech) for 3 days to obtain activated B cells. Immature B-cell populations were purified by FACs-sorting as follows: B220+ CD19−, pre-pro-B; B220+ CD19+ IgM-CD25− pro-B;

B220+ CD19+ IgM-CD25+ pre-B; B220+ CD19+ IgM+ IgD−, immature. Genomic DNA from purified B-cell populations was extracted, and PCR amplified with Pfu Ultra high-fidelity DNA polymerase (Stratagene). PCR reactions were fragmented with Covaris, and adapter-ligated libraries were generated, processed, and sequenced on a HiSeq2500 according to the manufacturer's instructions (Illumina).

**Flow cytometry, cell sorting, and cell viability assays.** The total mouse BM cells were obtained by washing the long bones with PBS with 1% FCS, using a 27-G needle. Cells were as well collected from peripheral blood, and also from the thymus and spleen after disrupting these organs by passing them through a 70-μm cell strainer. Erythrocytes were osmotically lysed using RCLB buffer, and nucleated cells were then washed with PBS–1% FCS. Cells were stained with the appropriate antibodies against the indicated cellular markers for 20 min at 4 °C, washed once with PBS–1% FCS, and finally they were resuspended in PBS–1% FCS with 10 μg/mL propidium iodide (PI) to exclude dead cells during data acquisition; this was performed in an AccuriC6 Flow Cytometer, and data files were analyzed using Flowjo software. For this analysis, gates were set by employing the commonly used forward and perpendicular light-scattering properties of mouse hematopoietic cells, and the specific fluorescence of the staining dyes used [FITC, PE, PI, and APC excited at 488 nm (0.4 W) and 633 nm (30 mW), respectively]; an example of such a gating strategy is shown in Supplementary Fig. 14. The nonspecific binding of staining antibodies to the Fc receptors of immune cells was prevented by incubating the samples with anti-CD16/CD32 Fc-block solution (clone 2.4G2, cat. #553142, BD Biosciences) for 20 min at 4 °C, previously to the addition of the staining antibodies. For each sample tube, a minimum of 50,000 living (i.e., PI-negative) cells were acquired and analyzed.

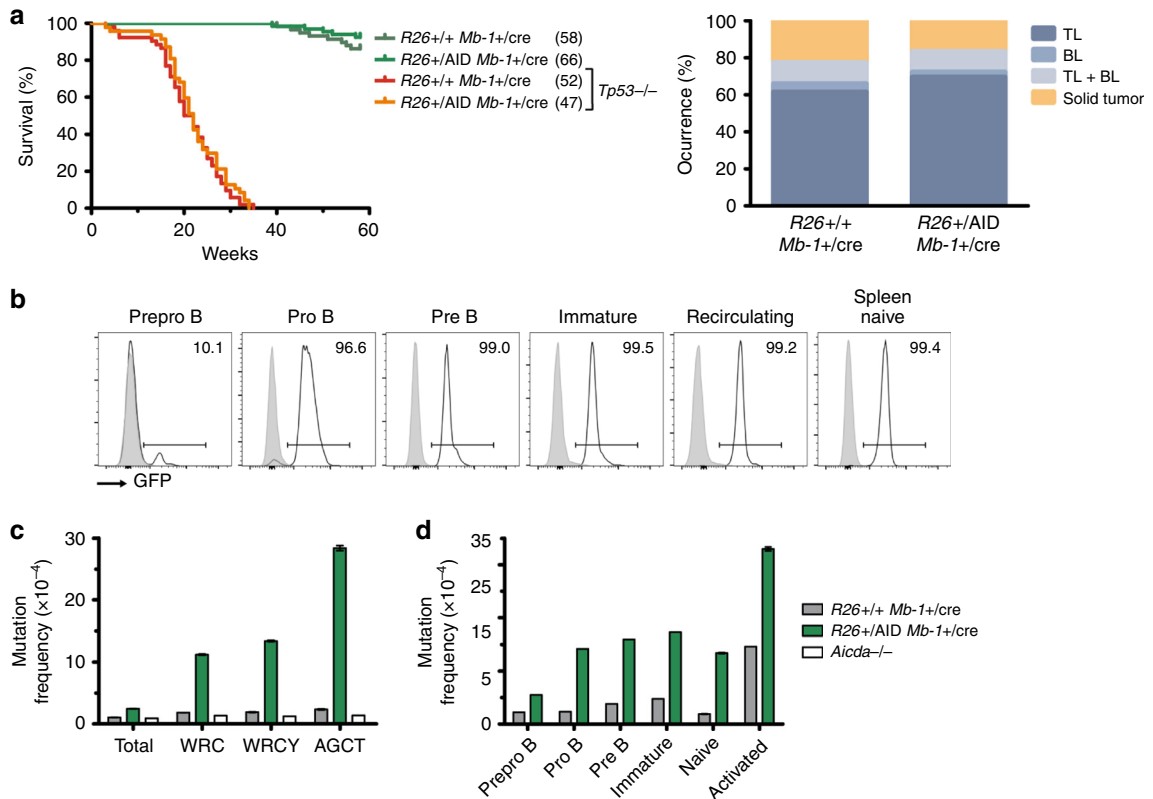

**Fig. 4 Early expression of AID in precursor B cells has no effect in lymphomagenesis. a** Kaplan–Meier survival graph for $R26^{+/AID}$ $Mb-1^{+/cre}$ and $R26^{+/+}$ $Mb-1^{+/cre}$ control mice in WT and $Tp53^{-/-}$ backgrounds. The number of mice in each group is indicated in brackets. Median survival for $R26^{+/AID}$ $Mb-1^{+/cre}$ $Tp53^{-/-}$, 22 weeks. Histopathology analysis of tumoral mice in $Tp53^{-/-}$ background. TL T lymphoma, BL B lymphoma. **b** GFP is fully expressed from the pro-B-cell stage on. GFP reporter expression in developing B cells of $R26^{+/AID}$ $Mb-1^{+/cre}$ mice (black line) and $Rosa26^{+/+}$ $Mb-1^{+/cre}$ control mice (gray shaded) in the bone marrow and naive B cells in the spleen. Populations were defined as B220+ CD19−, pre-pro-B; B220+ CD19+ IgM-CD25− pro-B; B220+ CD19+ IgM-CD25+ pre-B; B220+ CD19+ IgM+ IgD−, immature; B220+ CD19+ IgD+, recirculating; B220+, spleen naive. **c** SHM by R26-AID. SHM in Sμ region was analyzed by NGS in naive B cells purified from $R26^{+/AID}$ $Mb-1^{+/cre}$ and $Rosa26^{+/+}$ $Mb-1^{+/cre}$ control mice. $Aicda^{-/-}$ B cells were used as control for technical background. Total mutation frequency and specific mutation frequency in C or G within AID hotspots (WRC, WRCY, and AGCT) is indicated. Two mice per genotype and one $Aicda^{-/-}$ were analyzed. **d** SHM as measure of R26-AID activity in immature B-cell populations. Sorted B-cell populations from the bone marrow, purified naive, and LPS/IL4 activated B cells were used to analyze SHM in Sμ region as in **c**. Mutation frequency within WRCY AID hotspots is indicated. A pool of two mice per BM/naive populations and two independent mice for activated B cells were analyzed.

The antibodies used for flow cytometry were all from BD Biosciences: anti-B220 (clone RA3-6B2, cat. #103212, used in 1:100 dilution), CD4 (clone RM4-5, cat. #100516, used in 1:250 dilution), CD8a (clone 53-6.7, cat. #100708, used in 1:250 dilution), CD11b/Mac1 (clone M1/70, cat. #553310, used in 1:200 dilution), CD19 (clone 1D3, cat. #152404, used in 1:100 dilution), CD117/c-Kit (clone 2B8, cat. 105807, used in 1:200 dilution), Ly-6G/Gr1 (clone RB6-8C5; cat. #108412, used in 1:100 dilution), IgM (clone R6-60.2, cat. #406509, used in 1:100 dilution) and CD25 (clone PC61, cat. #553866, used in 1:100 dilution). The gating strategy is exemplified in Supplementary Fig. 14.

**Real-time PCR quantification (Q-PCR) of Aicda**. Expression of *Aid* was analyzed in sorted BM precursor B (B220low IgM−) cells of *wild-type*, *Pax5-het*, and *Sca1-ETV6-RUNX1* mice in two environmental conditions (mice housed in conventional facility and mice housed in an SPF facility) by Q-PCR as follows. cDNA for use in quantitative PCR studies was synthesized using reverse transcriptase (Access RT-PCR System; Promega, Madison, WI, USA). Real-time PCR reactions were performed in an Eppendorf MasterCycler Realplex machine. Assays used for quantitative PCR are commercially available from IDT (Integrated DNA Technologies): AID (Assay ID: Mm.PT.58.42247522) and GAPDH (Assay ID: Mm.PT.39a.1). In addition, the probes were designed so that genomic DNA would not be detected during the PCR. Measurement of GAPDH gene product expression was used as an endogenous control. The total spleen from immunized mice was used as a positive control of Aicda expression, and sorted BM pro/pre cells from *Aid-KO* mice were used as a negative control. All samples were run in triplicate. The comparative CT Method (ΔΔCt) was used to calculate relative expression of the transcript of interest and a positive control. The change in threshold cycle (ΔCt) of each sample was calculated as the Ct value of the tested gene (target) minus the Ct value of GAPDH (endogenous control). The ΔΔCt of each sample was obtained by

subtracting the ΔCt value of the reference from the ΔCt value of the sample. The ΔCt reference value used was the ΔCt obtained from the total spleen of wild-type immunized mice. The fold change in each group, calculated as $2^{-\Delta\Delta Ct}$ sample, was compared.

**In vitro Aid expression in preleukemic pro-B cells**. Pro-B cells were obtained from the BM of 2–3 weeks old WT, *Sca1-ETV6-RUNX1* and *Pax5-het* mice, respectively. B cells were isolated from the BM via positive MACS separation for B220. Cells were cultivated in special pro-B-cell medium (IMDM medium + L-glutamine + HEPES, 2% FCS, 1 mM penicillin–streptomycin, 0.03% Primaton, 50 μM beta-mercaptoethanol, 2 μg/ml cyproxin) supplemented with 5 ng/ml murine recombinant IL7 on ST2-feeder cells. Subsequently, pro-B cells were stimulated with TLR stimuli (LPS 40 μg/ml, PIC 25 μg/ml, R848 3 μg/ml, CpG (ODN 1668) 6 μg/ml; all from the company invivo Gen) and RT-PCR was performed for analysis of *Aid* expression (forward primer: GGGTCGTGAATGATGCTCTT, reverse primer: TGGCTTGTGATTGCTCAGAC).

**Histology**. Mice were euthanized by cervical dislocation. The tissue samples obtained after dissection were fixed in formaline and then included in paraffin following standard procedures. Hematoxylin–eosin stainings of representative sections were used for histopathology evaluation, performed under the supervision of Dr. Oscar Blanco, professional pathologist of human samples at the Salamanca University Hospital. For $Tp53^{-/-}$ background tumoral occurrence, lymphoid tissues were evaluated by histology, immunohistochemistry, and flow cytometry. Animals presenting tumors were classified into four categories according to the presence of solid tumors only or B/T-lymphomas independently of co-occurrence of additional solid tumors.

**Table 2 List of primers used to amplify immunoglobulin rearrangements.**

| | | |
|---|---|---|
| V$_H$J558 | Forward | CGAGCTCTCCARCACAGCCTWCATGCARCTCARC |
| | Reverse | GTCTAGATTCTCACAAGAGTCCGATAGACCCTGG |
| V$_H$7183 | Forward | CGGTACCAAGAASAMCCTGTWCCTGCAAATGASC |
| | Reverse | GTCTAGATTCTCACAAGAGTCCGATAGACCCTGG |
| V$_H$Q52 | Forward | CGGTACCAGACTGARCATCASCAAGGACAAYTCC |
| | Reverse | GTCTAGATTCTCACAAGAGTCCGATAGACCCTGG |
| DH | Forward | TTCAAAGCACAATGCCTGGCT |
| | Reverse | GTCTAGATTCTCACAAGAGTCCGATAGACCCTGG |
| Cμ | Forward | TGGCCATGGGCTGCCTAGCCCGGGACTT |
| | Reverse | GCCTGACTGAGCTCACACAAGGAGGA |

**V(D)J recombination**. The recombination of the immunoglobulin genes was analyzed by a PCR protocol based on the amplification of the possible products of V(D)J rearrangements based on the use of a primer immediately downstream of the J$_H$4 segment in several combinations with 5′ primers specific for the different families of V$_H$ segments. The primers are described in Table 2. PCR conditions were denaturation at 95 °C for 5 min, followed by 31–37 cycles of [(95 °C, 60 s) + (65 °C, 60 s) + (72 °C, 105 s)], followed by a final elongation step of 70 °C, 10 min. Then, PCR products were run on an agarose gel, and individual bands corresponding to specific rearrangements were isolated, cloned into the pGEM-T-easy vector (Promega), and sequenced; at least ten independent clones were sequenced for each individual band analyzed in order to determine the potential clonality of the rearrangement.

We have sequenced a selection of the PCR clonal bands, and have analyzed the V(D)J junction sequences with both IMGT/V-QUEST (http://www.imgt.org/)[52] and IgBLAST (www.ncbi.nlm.nih.gov/igblast)[53]. The information related to the primers used, band sizes, and the corresponding sequences are indicated in Supplementary Table 5.

**Microarray data analysis**. The total RNA was first isolated using TRIzol (Life Technologies), and then it was subjected to purification with the RNeasy Mini Kit (Qiagen) using also the On-Column DNase treatment option. Quality and quantification of RNA samples were determined by electrophoresis.

Determination of the expression of the different genes in the RNA samples was performed using Affymetrix Mouse Gene 1.0 ST arrays. All bioinformatic analyses of the array data were performed using R[54] and Bioconductor[55]. First, we applied background correction, intra- and inter-microarray normalization, and expression signal calculation using the microarray analysis algorithm[56–58], in order to determine the absolute expression signal for each gene in each array. Then, we used the significance analysis of microarray (SAM)[59] method to identify the gene probe sets with differential expression between experimental and control samples, SAM uses a permutation algorithm to allow to statistically infer the significance of the differential expression, and it provides P-values adjusted to correct for the multiple testing problem, by using FDR[60]. An FDR cutoff of <0.05 was used as a threshold to determine differential expression.

**Enrichment analysis**. In order to identify potential signatures of gene expression associated with different biological processes, gene set enrichment analysis (GSEA) was performed using the MSigDB databases from the Broad Institute[28,61,62]. Gene expression signatures that have been previously found to be specifically deregulated in human B-ALLs[28,62,63] were also analyzed to determine their potential overlap with the ones that were found up- or downregulated by the GSEA analysis of our tumor samples.

**Mouse exome library preparation and NGS**. DNA was purified from samples using the AllPrep DNA/RNA Mini Kit (Qiagen) according to the manufacturer's instructions. The exome library was prepared using the Agilent SureSelectXT Mouse All Exon Kit with some modifications. Exome capture was performed by hybridization to an RNA library according to the manufacturer's protocol. Then, the captured library was purified and enriched by binding to MyOne Streptavidin T1 Dynabeads (Life Technologies) and posterior off-bead PCR amplification in the linear range. Sequencing (2 × 100 bp) was carried out in a HiSeq2500 (Illumina) using the TruSeq SBS Kit v3 with a 6-bp index read.

**Data analysis**. Fastq files were generated with Illumina BcltoFastq 1.8.4. The alignment of the sequence data to the GRCm38.71 mouse reference genome was performed with BWA version 0.7.4. SAMtools was used for conversion steps and removal of duplicate reads. GATK 2.4.9 was used for local realignment around indels, SNP-calling, annotation, and recalibration. For recalibration, mouse dbSNP138 and dbSNP for the used mouse strains were used as training data sets. The variation calls obtained in this way were then annotated using the v70 Ensembl database with variant effect predictor (VEP), incorporating loss-of-function prediction scores for PolyPhen2 and SIFT. Afterward, the information was imported into an in-house MySQL database for further annotation, reconciliation, and data analysis by complex database queries if required.

Somatic calls were the output from MuTect[64] and VarScan[65]. For VarScan2 results, false-positive filtering was used as indicated by the author. In order to increase the reliability of the results, only calls having at least a 9% difference in allele frequency between tumor and normal samples were considered for further analysis. Cancer-related genes were singled-out by using the information from the Catalogue of Somatic Mutations in Cancer (COSMIC)[66] after having translated the cancer gene consensus from COSMIC by making use of Ensembl's BioMart[67].

**Sequencing**. Classical Sanger sequencing was used for the validation of mutations, using a 3130 Genetic Analyzer (Applied Biosystems). The sequencing primers used were: *mJak3* forward: CGGGATGTGGGGGCTTTAACT, reverse: GCAGACACGG GGTATAGTGG; *mPax5* forward: CTCGTACATGCACGGAGACA, reverse: GGA CCCTTCAGTACACCAGC.

**Pro-B-cell culture**. Pro-B cells were purified from BM using magnetic-activated cell sorting, selecting with anti-B220 beads (Milteny Biotec). Pro-B cells were maintained and expanded by culturing them in Iscove's Modified Dulbecco's Medium (IMDM) supplemented with 50 μM β-mercaptoethanol, 1 mM L-Gln, 2% heat-inactivated FCS, 1 mM penicillin–streptomycin (BioWhittaker), 0.03% (w/v) primatone RL (Sigma), and 5 ng/ml mrIL7 (R&D Systems), in the presence of Mitomycin C-treated ST2-feeder cells. Tumor pro-B cells that could grow independently of IL7 were grown in the same medium without this cytokine.

**Transplantation**. IL7-independent *Pax5-het/Aid-het* and *Pax5-het/Aid-KO* pro-B cells were intravenously injected into 12-week-old male syngenic mice (C57BL/6× CBA) that had previously been sublethally irradiated (4 Gy). Leukemia development in the injected mice was followed by regular analysis of peripheral blood, until the moment when leukemic blasts were detected in the blood; at this point, animals were killed and analyzed to characterize B-ALL development.

**Mutational pattern analysis in B-ALL**. Samples were subjected to a standardized alignment with [BWA-mem], and a preprocessing step consisting of [GATK realignment, recalibration, trimming tool single-nucleotide variants were identified for *Pax5-het/Aid-het* and *Pax5-het/Aid-KO* B-ALL and control samples with Platypus version 0.8.1 and default parameters[68]. Control variants were pooled and removed from each tumor sample's variant list, and vice versa. Furthermore, all variants with a quality score < 100 were discarded. Mutational signature analyses were subsequently conducted with the R/Bioconductor package MutationalPatterns, version 1.6.1[69].

**Statistical analysis**. The comparisons for the average values between two sample groups were made by two-tailed Student's t test or Wilcoxon's rank-sum test with GraphPad Prism software or R software. Mice survival was analyzed using Kaplan–Meier representation of the different cohorts, and statistics for differential survival were performed using the log-rank (Mantel–Cox) test. For in vivo transplantation experiments, the minimal number of mice in each group was calculated through use of the "cpower" function in the R/Hmisc package. The level of significance was set at $P < 0.05$.

**Reporting summary**. Further information on research design is available in the Nature Research Reporting Summary linked to this article.

## Data availability

Authors can confirm that all relevant data are included in the paper and/or its supplementary information files. All data reported in this article are deposited in the NCBI's Gene Expression Omnibus (GEO)[70], and are available under the GEO Series accession number GSE122105. The source data underlying Supplementary Fig. 6 is provided as a Source Data file.

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

## Acknowledgements

We would like to thank Dr. César Cobaleda for advice and help in the revision of the paper. We would also like to thank all members of our groups for useful suggestions and for their critical reading of the paper. Research in CVD group is partially supported by FEDER, "Miguel Servet" Grant (CP14/00082 - AES 2013–2016) from the Instituto de Salud Carlos III (Ministerio de Economía y Competitividad), "Fondo de Investigaciones Sanitarias/Instituto de Salud Carlos III" (PI17/00167). J.H. has been supported by ERAPer Med, the German Cancer AID (Translational Oncology Program 70112951), the DKTK German Cancer Consortium Joint funding program "Targeting MYC" L*10, The German Jose Carreras Foundation (DJCLS 02R/2016), and the Kinderkrebsstiftung (2016/17). A.R.R. was supported by ERC StG BCLYM-207844, SAF2013-42767-R and SAF2016-75511-R grants and co-funding by Fondo Europeo de Desarrollo Regional (FEDER). P.D. was funded by the Asociación Española Contra el Cancer. The CNIC is supported by the Ministerio de Ciencia, Innovación y Universidades (MCNU) and the Pro CNIC Foundation, and is a Severo Ochoa Center of Excellence (SEV-2015-0505). A.B. has been supported by the German Children's Cancer Foundation and the Federal Ministry of Education and Research, Bonn, Germany. Research in ISG group is partially supported by FEDER and by SAF2015-64420-R MINECO/FEDER, UE, RTI2018-093314-B-I00 MCIU/AEI/FEDER, UE, by Junta de Castilla y León (UIC-017, CSI001U16, and CSI234P18). I.S.G. lab is a member of the EuroSyStem and the DECIDE Network funded by the European Union under the FP7 program. A.B. and I.S.G. have been supported by the German Carreras Foundation (DJCLS R13/26) and by the German Federal Office for Radiation Protection (BfS)-Germany (FKZ: 3618S32274). G.R.H., A.C.-G and S.G.T.D. were supported by *FSE-Conserjería de Educación de la Junta de Castilla y León* (CSI001-15, CSI067-18 and CSI003-17, respectively). F.A. was supported by a Deutsche Forschungsgemeinschaft (DFG) fellowship (AU 525/1-1).

## Author contributions

Initial conception of the project was designed by J.H., I.S.-G., and A.B.; A.R.R., A.F.A.-P., and P.D. were responsible for the design, execution, and data interpretation of the AID gain-of-function mouse experiments; development of methodology was performed by G.R.-H., F.O., P.D., A.F. A.-P.; I.G.-H., J.R.-G., A.C.-G., A.O., O.B., D.A.-L., J.D.L.R., S.G.T.D., F.A., U.F., S.J., M.M., F.J.G.C., M.B.G.C., C.V.-D.; I.S.-G., and A.B.; O.B., M.B.G.C., F.J.G.C., and C.V.-D. performed pathology review; management of patient samples was performed by O.B., M.B.G.C., F.J.G.C., C.V.-D, J.H., A.B. G.R.-H., F.O., C.W., C.B., M.D., M.M., C.V.-D., I.S.-G., J.H., and A.B. were responsible for analysis and interpretation of the data (e.g., statistical analysis, biostatistics, computational analysis); paper preparation was performed by G.R.-H., F.O., P.D., A.F. A.-P.; I.G.-H., J.R.-G., A.C.-G., A.O., O.B., D.A.-L., J.D.L.R., S.G.T.D., C.W., C.B., M.D., F.A., U.F., S.J., M.M., F.J.G.C., M.B.G.C., C.V.-D., J.H., A.R.R.; I.S.-G., and A.B.; administrative, technical, or material support (i.e., reporting or organizing the data, constructing databases) was compiled by G.R.-H., F.O., C.V.-D., and I.S-G. The study was supervised by J.H., I.S.-G., and A.B.

## Competing interests

The authors declare no competing interests.
