## [Peer Review File · Nature Communications]

Reviewers' comments:

Reviewer #1 (Remarks to the Author):

This letter by Rodriguez-Hernandez and colleagues studies the role of Pax5, AID, and infection in the clonal evolution of B-lineage ALL. The authors find that approximately 25% of Pax5 het mice housed under "infectious" conditions develop B-ALL. The incidence of B-ALL was independent of AID genotype. The mutational spectra of the B-ALL samples was assessed by WES. The authors then enforced AID expression in pre-B cells, and found no increase in B-ALL; these experiments were performed in a Pax5 WT background. The authors then enforced expression of AID in a p53 -/- background and found no difference in onset nor type of malignancy. The authors conclude that although infectious stimuli can promote B cell transformation, the mechanism may be through AID independent mechanism.

The principal criticisms of the letter are: 1) relatively little new information; 2) controls for mutational analysis, and 3) choice of model systems. Pax5 deletions/mutations are most commonly associated with B cell precursor ALL thought to arise from B-cell precursors in the bone marrow. However, AID (*Aicda*) is principally restricted to splenic GC B cells, and not typically expressed in bone marrow B cell precursors; therefore, the rationale for combining these two lesions is unclear.

Major points:

- 1) No comparison to mice housed under SPF (ie, non-infectious) conditions was made in this study; the authors apparently relied upon historical controls. Are the authors confident that the infectious stimuli in the historical study is similar to their study? If so, why?
- 2) Figure 2a--The number of somatic SNV is remarkably high in some samples (833-2108). Were the tumor samples compared to germline tissue from the same mouse? What were the criteria for identifying SNV (VAF, # of mutant reads)? Did the authors verify that the mutations were indeed somatic? Until these questions are addressed, the data in figure 3 and 4c is difficult to interpret.
- 3) Figure 4c—Its not clear how the authors are able to identify mutants in polyclonal B cell precursors. One would expect AID-induced mutations in a pool of non-malignant cells to have very low VAF.

Minor points:

- 1) P. 4—where is the kit/CD25 staining shown?
- 2) Were the DJ/VDJ junctions sequenced? Were there any unique features regarding the junctions or choice of V, D, J segments?
- 3) The legend to figure 1b is confusing.

Reviewer #2 (Remarks to the Author):

NCOMMS-19-07109-T

Title:

Clonal evolution of pre-malignant B-cells in the absence of AID

In this work, Rodríguez-Hernandez and colleagues examined different approaches in order to understand if the mutagenic AID enzyme is required for clonal evolution of pre-malignant precursor B-cells in B-ALL. Their results are very interesting and could be controversial, taking into account the established idea, supported by numerous previous works, of AID-driven B-cell lymphomagenesis.

Nevertheless, the authors demonstrate that AID was dispensable for clonal evolution in a B-ALL mice model. To prove this, they performed loss-of-function experiments based on mice models overexpressing AID transgene and knocking-out this transgene. The authors evaluated the profile of mRNA expression in these models by arrays experiments and also performed WES in order to

visualize AID-driven mutagenesis. Finally, they extrapolated all these results to human leukemia using bioinformatics approaches. This work allowed the authors to reinforce their previous data, demonstrating that AID prevents pro-B ALL by functioning as a negative regulator, (Auer F., et al Oncotarget, 2017).

Despite the fact that, it is a well-designed and well-executed work, some concerns destined to improve and to clarify the central message of the manuscript must be addressed.

1) It has been clearly demonstrated that the majority of AID-initiated lesions occur near transcription start sites linked with multiple promoter and enhancer regions. As suggested by seminal works (Meng et al., Cell, 2014 and Quian et al., Cell, 2014), super-enhancer transcription converges on AID activity.

Although the work of Rodríguez-Hernandez et al. shows that AID off-target mutagenic activity in precursor B-cells does not promote B-ALL development, many other works have clearly showed the oncogenic activity of AID and their role in tumor development (Pasqualucci et al., Nature, 2001; Okasaki et al., JEM, 2003; Robbiani et al., Cell, 2015 and many others). It is logical to assume that a B-cell progenitor of a typical ALL has great differences concerning the DNA open transcriptional regions, regulatory transcriptional elements, AID accessibility, etc, compared with a mature and mutated neoplastic B-cell that has previously encountered its antigen in a germinal center. Taking into account that B-ALL is originated in a B-progenitor lineage, the authors should better discuss their results regarding the role that : i) differential chromatin remodelling, ii) different transcriptional activity (super-enhancers) and/or iii) AID DNA accessibility could have in tumour cells with diverse differentiation stages.

2) In the current work, evidence supporting that AID is dispensable for leukemogenesis was only formally provided for the case of murine Pax5-dependent B-ALL. Considering that ALL cases with Pax5 mutations account for about 7% of ALL cases, any generalization to other B-ALL subtypes should unequivocally be kept as speculative. This fact and limitation of the model should also be discussed.

3) Figure 1 should be revised regarding:

i) Figure 1b, upper panel, line for Aid-het (n=15) and for Aid-KO (n=15) are not depicted in the graph.

ii) Figure 1d, is not clear, and the legend is discordant with the image. The legend only mentions the comparison between GSEA of Pax5-het/Aid-KO and human BCR-ABL B-ALL gene set. In contrast, the picture states "leukemic Pax5-het + Aid-KO". What does Pax5-het + Aid-KO mean? Please clarify and improve this graph, the legend, and the labelling.

iii) Figure b-c and extended data fig.2: Despite the fact that the survival rate is similar between the 3 different models (Pax5-het/Aid-KO, Pax5-het/Aid-het and Pax5-het), in the figure 1c and in the extended data (figure 2) the flow cytometric analysis of bone marrow and peripheral blood show differences comparing the percentages of clonal tumor population between Pax5-het/Aid-het and Pax5-het/Aid-KO (95 % and 60 %, respectively). Given that in both figures the same animals (I142 and I921) are depicted, it would be desirable that the authors show a supplementary graph with statistical data from all animals. Pax5-het/Aid-het model (n=6) compared with Pax5-het/Aid-KO (n=6) in order to depict if significant differences exist or not. Despite the fact that survival and spleen infiltration appear to be similar in both models, if significant differences exist concerning this data, they will be should discussed.

4) In the same view, in the extended data (figure 6 and 7) a comparison of gene expression profiles between the new strain models (Pax5-het/Aid-KO and Pax5-het/Aid-het) should be performed and included. Data visualization of table I and table II depict some interesting genes

that are differentially expressed between the two models compared with their wt counterpart. Key genes involving immune response signalling and B cell activation, such as chemokines (Cxcl12, Cxcl14, Cxcl3, Cxcr4 etc); interleukins (IL-1, IL-6, IL-10 or their receptors) and Ikaros 1, 2, 3 and 5, appear to be differentially expressed. Similarly, tumor development genes such as Myc, Pten, Wnt5a and 8, Pxn, Foxo1 and Foxo3, are also overexpressed in the Pax5-het/AID-het compared with the Pax5-het/Aid-KO counterpart. On the other hand, a different quantity of miRs number is visualized comparing Pax5-het/AID-het (n=34) and Pax5-het/Aid-KO (n=3). The authors should discuss these data, if decided to present them into the manuscript.

Finally, figures 6 and 7 of extended data should be clarified, for example: What blue and red highlighted genes mean? How and why these genes were selected? Are these genes differentially expressed from others? Please clarify these questions at least in the figure legend. Names of the dendograms are missing.

5) Last paragraph of page 4 and first of page 5 involving figure 1d, present some concerns for this reviewer.

The authors state:

"... to explore the relevance of our findings in human leukemia, we next identified differentially expressed genes (FDR=0.01) in tumor-bearing BM from both Pax5-het/Aid-het (Extended Data Fig. 6 and Extended Table I) and Pax5-het/Aid-KO (Extended Data Fig. 7, Extended Table II) mice compared with BM precursor B cells from wild-type mice, showing that AID deficient B-ALL exhibit a gene expression pattern that is similar to human B-ALL18-20 (GSEA FDR=0.001; FDR= 0.000; FDR= 0.005; FDR= 0.002) (Fig. 1d)".

In this case mRNA arrays expression of BM precursor B cells of the generated models (Pax5-het/Aid-het and Pax5-het/Aid-KO) with human data of B-ALL should be reanalyzed or much better explained.

i) For this reviewer, to really explore the relevance of their data in the human B-ALL context, the authors should specifically evaluate the expression profile of mRNA from B-ALLs samples with higher AID expression (Philadelphia chromosome positive) and compare this profile with arrays performed on human B-ALLs with lower AID expression (Philadelphia chromosome negative), according to what Guangquan Zhou et al. described, (LEUKEMIA & LYMPHOMA, 2017). This data may be available in the study of Juric et al. (JCO, 2007). In this manuscript Juric et al., analyse human B-ALL taking into account BCR-ABL positive and negative samples, this could be an important input to specifically compare the role of AID activity on mRNA expression derived from tumor cells from patients with B-ALL expressing or not AID. It is not clear and not stated in the figure legend, if this analysis was performed.

Comparison of mRNA expression profile of the different mice models (Pax5-het/Aid-het and Pax5-het/Aid-KO) with previously reported arrays of unselected human B-ALL (described in Chiaretti et al., Clin Cancer Res, 2005 or Kohlmann et al., Leukemia, 2004) seems to be less pertinent.

6) Concerning WES analysis:

i) Which are the SNV named as somatic cancer? It is probably the most relevant information taking into account the initial hypothesis concerning AID activity on specific off-target genes. Furthermore, there seem to be some differences here, at least taking into account the number of mutated somatic cancer genes in Pax5-het/Aid-het model (n=48) compared with the Pax5-het/Aid-KO model (n=19). A supplementary table showing these genes and the mutational context could be included (highlighting for example canonical and non-canonical AID hotspots).

ii) The authors state ...“we highlighted these present SNVs and could show that”.... How did the authors “highlighted” these genes? Which are the selection criteria? Please clarify these points. Furthermore, for specific genes that appear in the Pax5-het/Aid-het model such as Jak3, Fancm and others, the authors should depict or mention if the mutations are located in AID hotspots or not. These data could help to support previous information showing that AID is functional in the model.

7) The final paragraph referencing the fact that AID levels could be increased by PI3Kdelta of Bruton's tyrosine kinase inhibitors should be more rigorously stated and updated. The manuscript of Campagno et al. shows increased levels of AID only after drugs treatment in-vitro and in tumor cell lines. Furthermore, a significant AID activity on DNA of treated patients was only demonstrated for PI3Kdelta, not for ibrutinib. If authors decide to maintain this paragraph, they should change it updating the information and consider the evidence in a recently published work by Morande et al., Blood, 2019.

8) Significant references for the message of this manuscript such as Jinjung Dang et al., Blood 2015 and Guangquan Zhou et al., LEUKEMIA & LYMPHOMA, 2017, should be incorporated.

Point-by-point response:

Referee #1:

“This letter by Rodriguez-Hernandez and colleagues studies the role of Pax5, AID, and infection in the clonal evolution of B-lineage ALL. The authors find that approximately 25% of Pax5 het mice housed under “infectious” conditions develop B-ALL. The incidence of B-ALL was independent of AID genotype. The mutational spectra of the B-ALL samples was assessed by WES. The authors then enforced AID expression in pre-B cells, and found no increase in B-ALL; these experiments were performed in a Pax5 WT background. The authors then enforced expression of AID in a p53 -/- background and found no difference in onset nor type of malignancy. The authors conclude that although infectious stimuli can promote B cell transformation, the mechanism may be through AID independent mechanism. The principal criticisms of the letter are: 1) relatively little new information; 2) controls for mutational analysis, and 3) choice of model systems. Pax5 deletions/mutations are most commonly associated with B cell precursor ALL thought to arise from B-cell precursors in the bone marrow. However, AID (Aicda) is principally restricted to splenic GC B cells, and not typically expressed in bone marrow B cell precursors; therefore, the rationale for combining these two lesions is unclear.”

We thank the reviewer for his/her thoughtful assessment of our manuscript and the constructive comments and suggestions for improving it. Due to the necessary shortening of the introductory part the rationale of combining the two investigated lesions (PAX5, AID) was not explained clearly by us. We revised this part of the manuscript (pages 2-3 containing the new Introduction section of the revised manuscript) and apologize for potential misunderstanding caused by this.

We completely agree with the reviewer that AID is usually expressed in splenic GC B cells and plays a well established role in B cell lymphomagenesis in this context. In contrast, AID is not generally expressed in early bone marrow B cell precursors, and therefore the link to leukemia development is not intuitive.

However, the current dogma in the field of B-cell leukemogenesis holds that AID expression is induced in pre-leukemic B-cell precursor cells in response to infection and promotes in this case secondary genetic changes that may lead to subsequent leukemia development. This model is presented in a figure of a recent review in Nature Reviews of Cancer (Mel Greaves, 2018 Aug;18(8):471-484), please see below.

[REDACTED]

However, evidence supporting this dogma has been largely acquired through the use of *ex vivo* functional studies involving bone marrow transplantation. Whether AID also contributes to native (non-transplant) B-ALL development is to date entirely unclear. In the present manuscript, we addressed this question using two novel genetic *in vivo* models: (1) an experimental mice model that develops B-ALL only in response to natural infection (the “*pax5-het*” model, which faithfully mimicks human disease with respect to disease penetrance, clinical and genomic characteristics), and (2) a complementary mouse model for conditional *Aid* expression in B-cell precursors.

Using these two models we clearly show that *ex vivo* approaches are not representative for the *in vivo* situation in this case. We demonstrate (1) that *Aid* expression levels are not upregulated in preleukemic precursor B cells in mice prone to develop B-ALL under natural infection exposure, (2) that genetically induced *Aid* expression and generation of *Aid* off-target mutations in bone marrow precursor B-cells does not promote B-ALL development and (3) that the presence or absence of AID does not impact on the latency or incidence of infection-mediated B-ALL development. This *Aid*-independently emerged B-ALL closely resembles the human disease in pathology, genomic lesions, and leukemia-associated transcripts. Our study therefore provides evidence for a substantially revised model of leukemogenesis triggered by infection.

Referee 1 also indicated, quite rightly, some points that need clarification. We have carefully addressed all of the comments in the revised manuscript as detailed below:

Major points:

1) No comparison to mice housed under SPF (ie, non-infectious) conditions was made in this study; the authors apparently relied upon historical controls. Are the authors confident that the infectious stimuli in the historical study is similar to their study? If so, why?

We thank the reviewer for this well considered comment. A comparison to mice housed under SPF was not made in the current study, because we focus on the mechanisms of leukemogenesis and these animals never develop leukemia. At all times, also control mice are housed under SPF. Please, find below a summary of all corresponding mice models kept in SPF and conventional facility (CF) by us and the respective occurrence of B-cell ALL in these cohorts.

	Mice housed in SPF facility		Mice housed in CF facility	
	Total number of mice in the study	Mice with B-ALL	Total number of mice in the study	Mice with B-ALL
Aid-het	83	0	15	0
Aid-KO	76	0	15	0
Pax5-het	16	0	41	9
Pax5-het/Aid-het	19	0	30	9
Pax5-het/Aid-KO	37	0	23	7
Sum	231	0	124	25

The stimulating infectious environment of the mice is not a subject of change between studies as evidenced and tested by the following two measures:

- 1) **The rate of leukemia development is constant:** As sentinels, we permanently keep Pax5-het mice in the SPF and the conventional facility. These Pax5-het mice never developed B-ALL in SPF, but a constant rate of 25% of mice developed leukemia in the conventional facility indicating the presence of similar infectious stimuli in the conventional facility over time.
- 2) **The microbiological status in our SPF and conventional facilities is well defined, closely monitored and has not changed since the earlier studies in 2015.** In the revised version of the manuscript we now include two new tables (New Extended Data Tables I and II) listing the pathogens tested to monitor the health status of the animals housed in both facilities (SPF and conventional facility). From 2015 (corresponding to the historical study) to 2019, no change in the monitored pathogen exposition has been observed.

2) *Figure 2a--The number of somatic SNV is remarkably high in some samples (833-2108). Were the tumor samples compared to germline tissue from the same mouse? What were the criteria for identifying SNV (VAF, # of mutant reads)? Did the authors verify that the mutations were indeed somatic? Until these questions are addressed, the data in figure 3 and 4c is difficult to interpret.*

We thank the reviewer for this comment. Indeed, two of the analyzed mice exhibited a comparatively high amount of somatic variations reminiscent of mutator phenotypes of individuals with DNA repair defects. As the underlying genetic reason is unclear we omitted these samples from our analyses. To increase the cohort size we sequenced additional mice and updated **Figure 2a and 2b** accordingly.

As described on page 5 of the original manuscript, to identify somatic variations we used whole exome sequencing and compared matched tumor and germline tissue of the same mice:

“...In order to further identify somatically acquired 2nd hits leading to leukemia development in both Pax5-het/Aid-het and Pax5-het/Aid-KO B-ALLs, we performed whole exome sequencing of 6 Pax5-het/Aid-het tumors, 6 Pax5-het/Aid-KO tumors and paired samples of germline DNA as a reference. ...”

To identify somatic calls we used the bioinformatic tools MuTect and VarScan and stringently filtered the data for false-positive calls as was described in the supplementary information (on page 10-11 of the original Supplemental information file):

“...Somatic calls were produced using MuTect and VarScan. For VarScan2 results, false-positive filtering as suggested by the author was applied. ..., only entries with at least 9% difference in allele frequency between tumor and normal were kept for further analysis....”

To generate **Figure 3** (mutational pattern analysis) the same whole exome sequencing data, but a different bioinformatic approach was used as was described in the supplementary information (on page 12-13 of the original Supplemental information file):

“Samples were subjected to a standardized alignment with [BWA-mem], and a preprocessing step consisting of [GATK realignment, recalibration, trimming tool Single nucleotide variants were identified for tumor and control samples with Platypus version 0.8.1 and default parameters. Control variants were pooled and removed from each tumor sample's variant list, and vice versa. Furthermore, all variants with a quality score < 100 were discarded. Mutational signature analyses were subsequently conducted with the R/Bioconductor package “MutationalPatterns”, version 1.6.1 27.”

We recalculated the results omitting the two datasets with very high mutation rates and including new sequencing data from additional mice as explained above and updated Figure 3. The median number of mutations was 524 for the whole group, the correlation "real vs replicated" for the profiles was 0.967 for de novo decomposition and 0.943 for COSMIC decomposition.

The obtained result remained the same. Neither the *de novo* nor the COSMIC signatures revealed any group differences between Aid wt, +/- or -/- mice.

In the case of **Figure 4c** a very different type of data is shown: mutation frequency (rather than VAF) using a different type of approach (PCRseq) and on a different mouse model (R26+/AID Mb1Creki/+), as explained in more detail in the next paragraph.

3) Figure 4c—Its not clear how the authors are able to identify mutants in polyclonal B cell precursors. One would expect AID-induced mutations in a pool of non-malignant cells to have very low VAF.

We apologise for not having stressed more clearly that **Figure 4** is not based on whole exome sequencing data, but presents a different, independent approach in an additional complementary mouse model. In Figure 4c we are detecting AID-induced mutations in the R26+/AID Mb1Creki/+ mouse model, where we drive premature AID expression in early B cell precursors (Figure 4b). It is known that the Smu region -which preceeds the constant region in the immunoglobulin heavy chain locus- is a natural target for AID mutational activity. Mutations in the Smu region driven by AID are naturally non-clonal and can occur concomitantly with class switch recombination (Reina-San-Martin et al, J Exp Med. 2003 Jun 16;197(12):1767-78; Nagaoka et al, J. Exp. Med. 2002 Feb 18;195(4): 529–534). Therefore, analyzing mutations in the Smu region is a common way to track AID mutational activity. In Figure 4c we show that R26+/AID Mb1Creki/+ B cells accumulate AID-driven mutations in a natural AID target. Therefore, the function of the heterologous protein was not compromised. To measure AID-induced mutations in the Smu region we made use of PCR-Seq, an experimental approach previously developed in our lab (Perez-Duran et al, J Exp Med. 2012 Jul 2;209(7):1379-89), which allows high-depth, high sensitivity detection of AID-induced mutations. Thus, in Figure 4c and 4d, we show the mutation frequency found on the whole region, and on AID hotspots (WRC, WRCY, AGCT) within the region. As a background control, we show the mutation frequency in AID deficient cells. These values are in the same range as others previously reported for AID mutational activity in the Smu region (Reina-San-

Martin et al, Nagaoka et al, Perez-Duran et al). On page 5 of the original “Supplemental information file” we described that the analysis of mutations at S_μ by NGS was performed as described in Pérez-Durán P et al 2012.

Minor points:

1) P. 4—where is the kit/CD25 staining shown?

We apologize for this inadvertent omission, which we have now corrected. FACS data showing the kit/CD25 staining is included within the **New Extended Data Figure 2** in the revised version of the manuscript.

2) Were the DJ/VDJ junctions sequenced? Were there any unique features regarding the junctions or choice of V, D, J segments?

There are no specific features conditioning the choice of V, D, J segments. This is already suggested by the heterogeneity of the V and J segments identified by PCR in the clonal tumors shown in the Extended Data Fig. 6. Following the reviewer’s advice, we further analyzed if there are any other features relevant at the junction sequence level. We sequenced a selection of PCR clonal bands, and analyzed the V(D)J junction sequences using both IMGT/V-QUEST (<http://www.imgt.org/>) (Brochet et al., Nucl. Acids Res 2008) and IgBLAST (www.ncbi.nlm.nih.gov/igblast) (Ye et al., Nucl. Acids Res. 2013). Representative results of these analyses are shown in the **new Extended Data Fig. 7**, and the heterogeneity of the V, D and J junction sequences identified suggests that there is no selective pressure on the choice of the segments at the sequence level.

3) The legend to figure 1b is confusing.

We apologize for the inadvertent incorrect sentence in the legend. In the revised version, this sentence has been replaced by “*B-ALL–specific survival curve of Aid-het mice (light blue, n=15), Aid-KO mice (deep blue, n=15), Pax5-het/Aid-het mice (violet, n=23), Pax5-het/Aid-KO mice (green, n = 30), and Pax5-het animals (red, n = 41) is compared to survival of WT control mice (black, n=20). Pax5-het/Aid-het or Pax5-het/Aid-KO mice showed a significantly shortened lifespan (log-rank p-value = 0.0061) (upper panel), that was similar to the Pax5-het/Aid-wt group (lower panel)*”.

Referee #2:

In this work, Rodríguez-Hernandez and colleagues examined different approaches in order to understand if the mutagenic AID enzyme is required for clonal evolution of pre-malignant precursor B-cells in B-ALL. Their results are very interesting and could be controversial, taking into account the established idea, supported by numerous previous works, of AID-driven B-cell lymphomagenesis.

Nevertheless, the authors demonstrate that AID was dispensable for clonal evolution in a B-ALL mice model. To prove this, they performed loss-of-function experiments based on mice models overexpressing AID transgene and knocking-out this transgene. The authors evaluated the profile of mRNA expression in these models by arrays experiments and also performed WES in order to visualize AID-driven mutagenesis. Finally, they extrapolated all these results to human leukemia using bioinformatics approaches. This work allowed the authors to reinforce their previous data, demonstrating that AID prevents pro-B ALL by functioning as a negative regulator, (Auer F., et al Oncotarget, 2017).

Despite the fact that, it is a well-designed and well-executed work, some concerns destined to improve and to clarify the central message of the manuscript must be addressed.

We greatly appreciate this kind appraisal of our study by Reviewer #2, and the thoughtful comments. By carefully addressing all the comments we could significantly strengthen the revised manuscript as detailed below:

Comments

1) *It has been clearly demonstrated that the majority of AID-initiated lesions occur near transcription start sites linked with multiple promoter and enhancer regions. As suggested by seminal works (Meng et al., Cell, 2014 and Quian et al., Cell, 2014), super-enhancer transcription converges on AID activity. Although the work of Rodríguez-Hernandez et al. shows that AID off-target mutagenic activity in precursor B-cells does not promote B-ALL development, many other works have clearly showed the oncogenic activity of AID and their role in tumor development (Pasqualucci et al., Nature, 2001; Okasaki et al., JEM, 2003; Robbiani et al., Cell, 2015 and many others). It is logical to assume that a B-cell progenitor of a typical ALL has great differences concerning the DNA open transcriptional regions, regulatory transcriptional elements, AID accessibility, etc, compared with a mature and mutated neoplastic B-cell that has previously encountered its antigen in a germinal center. Taking into account that B-ALL is originated in a B-progenitor lineage, the authors should better discuss their results regarding the role that : i) differential chromatin remodelling, ii) different transcriptional activity (super-enhancers) and/or iii) AID DNA accessibility could have in tumour cells with diverse differentiation stages.*

We want to thank the reviewer for this valuable comment. We have now have added these aspects to the discussion to the revised version of the manuscript (pages 11-12).

2) *In the current work, evidence supporting that AID is dispensable for leukemogenesis was only formally provided for the case of murine Pax5-depquent B-ALL. Considering that ALL cases with Pax5 mutations account for about 7% of ALL cases, any generalization to other B-ALL subtypes should unequivocally be kept as speculative. This fact and limitation of the model should also be discussed.*

We thank the reviewer for this important critic. We previously stated in the abstract that “AID was dispensable for clonal evolution in this model”, but did not discuss this point further. In the revised version of the manuscript we have now more clearly explained both the importance of the model and but also the concurrent limitations in the discussion of the revised version of the manuscript (pages 11-12).

3) *Figure 1 should be revised regarding:*

i) *Figure 1b, upper panel, line for Aid-het (n=15) and for Aid-KO (n=15) are not depicted in the graph.*

The lines for Aid-het (n=15) and Aid-KO (n=15) are depicted in Figure 1b, but due to overlap with the line for WT mice they are not visible in the graph. We apologize for this misleading presentation. We now mention this in the legend of the respective Figure.

ii) *Figure 1d, is not clear, and the legend is discordant with the image. The legend only mentions the comparison between GSEA of Pax5-het/Aid-KO and human BCR-ABL B-ALL gene set. In contrast, the picture states "leukemic Pax5-het + Aid-KO". What does Pax5-het + Aid-KO mean? Please clarify and improve this graph, the legend, and the labelling.*

We apologize for the misleading picture. By mistake, we used “Pax5-het + Aid-KO” synonymously to Pax5-het/Aid-KO in the Figure labeling. In the revised version, we clarified and improved Figure 1d, its legend and its labeling.

iii) Figure b-c and extended data fig.2: Despite the fact that the survival rate is similar between the 3 different models (Pax5-het/Aid-KO, Pax5-het/Aid-het and Pax5-het), in the figure 1c and in the extended data (figure 2) the flow cytometric analysis of bone marrow and peripheral blood show differences comparing the percentages of clonal tumor population between Pax5-het/Aid-het and Pax5-het/Aid-KO (95 % and 60 %, respectively). Given that in both figures the same animals (I142 and I921) are depicted, it would be desirable that the authors show a supplementary graph with statistical data from all animals. Pax5-het/Aid-het model (n=6) compared with Pax5-het/Aid-KO (n=6) in order to depict if significant differences exist or not. Despite the fact that survival and spleen infiltration appear to be similar in both models, if significant differences exist concerning this data, they will be should discussed.

We thank the reviewer for this valuable suggestion. We have now have added a **new Extended Data Figure 5** with statistical data from all animals. The data shows no significant statistical differences between Pax5-het/Aid-het and Pax5-het/Aid-KO concerning e.g. the percentages of blast cells in BM, spleen and PB. We have also added a respective paragraph to the results section of the revised version of the manuscript (page 6).

4) In the same view, in the extended data (figure 6 and 7) a comparison of gene expression profiles between the new strain models (Pax5-het/Aid-KO and Pax5-het/Aid-het) should be performed and included. Data visualization of table I and table II depict some interesting genes that are differentially expressed between the two models compared with their wt counterpart. Key genes involving immune response signalling and B cell activation, such as chemokines (Cxcl12, Cxcl14, Cxcl3, Cxcr4 etc); interleukins (IL-1, IL-6, IL-10 or their receptors) and Ikaros 1, 2, 3 and 5, appear to be differentially expressed. Similarly, tumor development genes such as Myc, Pten, Wnt5a and 8, Pxn, Foxo1 and Foxo3, are also overexpressed in the Pax5-het/AID-het compared with the Pax5-het/Aid-KO counterpart. On the other hand, a different quantity of miRs number is visualized comparing Pax5-het/AID-het (n=34) and Pax5-het/Aid-KO (n=3). The authors should discuss these data, if decided to present them into the manuscript.

Following reviewer suggestion, we performed a new comparison of gene expression profiles of Pax5-het/Aid-KO B-ALL vs. Pax5-het/Aid-het B-ALL. There were no significant differences between these two groups. This result is now presented in page 6 and in **new Extended Data Figure 10** in the revised version of the manuscript.

Finally, figures 6 and 7 of extended data should be clarified, for example: What blue and red highlighted genes mean? How and why these genes were selected? Are these genes differentially expressed from others? Please clarify these questions at least in the figure legend. Names of the dendograms are missing.

In Figure 6 and 7 of the extended data upregulated genes are displayed in red and downregulated genes in blue. The genes shown are significantly induced or repressed as determined by significance analysis of microarrays using FDR 0.01%. We indicated this in the respective figure legend in the previous and current supplementary information: “Genes significantly induced or repressed as determined by significance analysis of microarrays using FDR 0.01%. Each row represents a separate gene, and each column denotes a separate mRNA sample. The level of expression of each gene in each sample is represented using a red–blue color scale (upregulated genes are displayed in red and downregulated genes in blue)”. In the revised version we included the names of the dendograms and the reason why some genes are highlighted.

5) Last paragraph of page 4 and first of page 5 involving figure 1d, present some concerns for this reviewer. The authors state:

"... to explore the relevance of our findings in human leukemia, we next identified differentially expressed genes (FDR=0.01) in tumor-bearing BM from both Pax5-het/Aid-het (Extended Data Fig. 6 and Extended Table I) and Pax5-het/Aid-KO (Extended Data Fig. 7, Extended Table II) mice compared with BM precursor B cells from wild-type mice, showing that AID deficient B-ALL exhibit a gene expression pattern that is similar to human B-ALL18-20 (GSEA FDR=0.001; FDR= 0.000; FDR= 0.005; FDR= 0.002) (Fig. 1d)".

In this case mRNA arrays expression of BM precursor B cells of the generated models (Pax5-het/Aid-het and Pax5-het/Aid-KO) with human data of B-ALL should be reanalyzed or much better explained.

This comment is addressed below.

i) For this reviewer, to really explore the relevance of their data in the human B-ALL context, the authors should specifically evaluate the expression profile of mRNA from B-ALLs samples with higher AID expression (Philadelphia chromosome positive) and compare this profile with arrays performed on human B-ALLs with lower AID expression (Philadelphia chromosome negative), according to what Guangquan Zhou et al. described, (LEUKEMIA & LYMPHOMA, 2017). This data may be available in the study of Juric et al. (JCO, 2007). In this manuscript Juric et al., analyse human B-ALL taking into account BCR-ABL positive and negative samples, this could be an important input to specifically compare the role of AID activity on mRNA expression derived from tumor cells from patients with B-ALL expressing or not AID. It is not clear and not stated in the figure legend, if this analysis was performed.

Comparison of mRNA expression profile of the different mice models (Pax5-het/Aid-het and Pax5-het/Aid-KO) with previously reported arrays of unselected human B-ALL (described in Chiaretti et al., Clin Cancer Res, 2005 or Kohlmann et al., Leukemia, 2004) seems to be less pertinent.

We want to thank the reviewer for this valuable suggestion and completely agree with his/her view that a comparison of mRNA expression between human B-ALL cells expressing high or low levels of AID would be a true reality check with the human context. Previously we compared the expression profiles of Pax5-het/Aid-KO B-ALL with the data available in the study of Juric et al. (*J Clin Oncol* **25**, 1341-1349, 2007) (Figure 1d). We have now performed the new analyses suggested by this reviewer and added the results as a **novel Figure 1e**. A respective paragraph was added to the results section of the revised manuscript. In accordance with the lack of significantly different gene expression profiles between Pax5-het/Aid-KO B-ALL and Pax5-het/Aid-het B-ALL (new Extended Data Figure 10) our study showed no significant (FDR>0.25) enrichment of human BCR/ABL B-ALL genesets (extracted from *J Clin Oncol* **25**, 1341-1349, 2007), between leukemic Pax5-het/Aid-het and leukemic Pax5-het/Aid-KO phenotypes (pages 6-7).

6) Concerning WES analysis:

i) Which are the SNV named as somatic cancer? It is probably the most relevant information taking into account the initial hypothesis concerning AID activity on specific off-target genes. Furthermore, there seem to be some differences here, at least taking into account the number of mutated somatic cancer genes in Pax5-het/Aid-het model (n=48) compared with the Pax5-het/Aid-KO model (n=19). A supplementary table showing these genes and the mutational context could be included (highlighting for example canonical and non-canonical AID hotspots).

We thank the reviewer for this valuable comment. The SNVs named “somatic cancer SNVs” are affecting genes known to play an important role in cancer and are listed in the cancer gene consensus database (Ref: *Nat Rev Cancer*, 2004 Mar;4(3):177-83., *A census of human cancer genes*, Futreal et al.).

There is no significant difference in the number of mutated somatic cancer genes between the *Pax5-het/Aid-het* and *Pax5-het/Aid-KO* tumors. Two tumors (I142T, *Pax5-het/Aid-het* and I520T *Pax5-het/Aid-KO*, respectively) displayed very high numbers of mutations (38 and 14, respectively) presumably due to mutations in DNA repair pathway genes. These skewed the results and were therefore removed from the analyses due to a comment of reviewer 1.

As suggested by this reviewer, in the revised version we include a new supplementary table (**new Extended Data Table V**) showing the mutated genes and the mutational context. In this table we indicate whether the genes are located in AID hotspots or not.

ii) The authors state ...“we highlighted these present SNVs and could show that”.... How did the authors "highlighted" these genes? Which are the selection criteria? Please clarify these points. Furthermore, for specific genes that appear in the Pax5-het/Aid-het model such as Jak3, Fancm and others, the authors should depict or mention if the mutations are located in AID hotspots or not. These data could help to support previous information showing that AID is functional in the model.

We thank the reviewer for this important comment. The SNVs selected are somatic and affecting genes with known oncogenic potential (referred to by us as “somatic cancer SNVs”). They are listed in the cancer gene consensus database. As suggested by the reviewer, in the revised version we include a supplementary table (**new Extended Data Table V**) showing these mutated genes and the mutational context highlighting whether they are located in AID hotspots or not.

7) The final paragraph referencing the fact that AID levels could be increased by PI3Kdelta of Bruton's tyrosine kinase inhibitors should be more rigorously stated and updated. The manuscript of Campagno et al. shows increased levels of AID only after drugs treatment in-vitro and in tumor cell lines. Furthermore, a significant AID activity on DNA of treated patients was only demonstrated for PI3Kdelta, not for ibrutinib. If authors decide to maintain this paragraph, they should change it updating the information and consider the evidence in a recently published work by Morande et al., Blood, 2019.

We thank the reviewer for this constructive comment. Following his/her advice, the last paragraph has been rewritten and the very recently published work by Morande et al., Blood, 2019 was included.

8) Significant references for the message of this manuscript such as Jinjung Dang et al., Blood 2015 and Guangquan Zhou et al., LEUKEMIA & LYMPHOMA, 2017, should be incorporated.

Following the reviewer's advice, these references are now discussed and included as references 5 and 24 in the revised version of the manuscript.

Reviewers' comments:

Reviewer #1 (Remarks to the Author):

The revised manuscript by Rodriguez-Hernandez et al have addressed several, but not all of my prior comments. The most fundamental shortcoming is that this is basically a negative study, showing that AICDA expression has fairly little effect on the generation of Pax5 induced B cell precursor ALL. While the role of AICDA in germinal center and post germinal center B cell malignancy is well established, and supported by numerous publications, I don't think it is commonly accepted "dogma" that AICDA plays an important role in B cell precursor ALL. Dr. Greaves is certainly an expert in the field, but his proposed model lists no published references supporting a role of AICDA in B cell precursor ALL. A recent review of B cell precursor ALL in the NEJM (Hunger and Mullighan, 2015) does not mention AICDA as an important driver for B cell precursor ALL. And the Immgen database (<http://www.immgen.org/databrowser/index.html>); search criteria "key Populations" and "aicda" shows that Aicda is not expressed in B cell precursors. This is supported by the authors figure 1, which also shows B cell precursors do not express Aicda.

Taking all of these features into consideration, I do not think it is surprising that Aicda knockout has little effect on B cell precursor ALL. Post germinal center lymphoma are not equivalent to B cell precursor ALL.

The analysis of WES data remains confusing. The authors compare matched germline and tumor, but find more variants in the germline than the tumor. Does this mean that the tumors have net negative acquired somatic changes? Or perhaps the filtering criteria are not adequately strict.

Reviewer #2 (Remarks to the Author):

The authors have addressed my comments and significantly strengthened the manuscript.

Point-by-point response:

Referee #1:

The revised manuscript by Rodriguez-Hernandez et al have addressed several, but not all of my prior comments.

We thank the reviewer for judging several of his/her comments to be addressed by us.

The most fundamental shortcoming is that this is basically a negative study, showing that AICDA expression has fairly little effect on the generation of Pax5 induced B cell precursor ALL. While the role of AICDA in germinal center and post germinal center B cell malignancy is well established, and supported by numerous publication, I don't think it is commonly accepted "dogma" that AICDA plays an important role in B cell precursor ALL. Dr. Greaves is certainly an expert in the field, but his proposed model lists no published references supporting a role of AICDA in B cell precursor ALL.

Dr. Greaves' notion (Greaves M. Nat Rev Cancer 18, 471-484, 2018) that AICDA plays an important role in B cell precursor ALL development is suggested by several lines of research and would provide not only a plausible, long sought answer to the question what causes the secondary genetic lesions leading to childhood leukemia but would also present a possible target for treatment. Dr. Greaves refers to several publications in his review that were mentioned in Box 2 and the review text (please see below):

[REDACTED]

Further references:

Olinski, R., Styczynski, J., Olinska, E. & Gackowski, D. Viral infection- oxidative stress/DNA damage- aberrant DNA methylation: separate or interrelated events responsible for genetic instability and childhood ALL development? **Biochim. Biophys. Acta** 1846, 226–231 (2014).

Swaminathan S, Muschen M. Infectious origins of childhood leukemia. **Oncotarget** 6, 16798-16799 (2015).

Greaves M, Muschen M. Infection and the Perils of B-cell Activation. **Cancer Discov** 5, 1244-1246 (2015).

To our knowledge the debate arose in 2008 with a publication by Tsai AG et al. in **Cell** (2008 Dec 12;135(6):1130-42).

We cite from a review in Cell by Grace K. Mahowald, Jason M. Baron, and Barry P. Sleckman: Collateral damage from antigen receptor gene diversification. *Cell*. 2008, DOI: 10.1016/j.cell.2008.10.035:

...”From these findings, it is proposed that AID may modify methylated CpG sequences, permitting the formation of T:G mismatches that, in turn, form substrates for RAG-mediated cleavage. A major tenet of this proposal is that B cells can express both RAG and AID simultaneously. In this regard, Tsai et al. note that translocations involving CpG clusters are markedly enriched in human lymphoid malignancies that appear phenotypically to be derived from pro-B/pre-B cells. From this, they speculate that AID and RAG may both be expressed in some developing human B cells and can function together in these cases to generate off-target DSBs that participate in translocations. Although this notion is consistent with the expression of AID in developing B cells from a quasimonoclonal mouse (as a consequence of restricted heavy and light chain usage), mice with a sensitive AID expression reporter failed to reveal any AID expression in developing B cells (Crouch et al., 2007, Mao et al., 2004)...”

We hope that our study will finally demonstrate that AICDA is -as the reviewer suggests- not involved in preB-ALL development. We agree with the reviewer that the wording “dogma” might be too strong and replaced it in the text by “view” or “hypotheses”.

A recent review of B cell precursor ALL in the NEJM (Hunger and Mullighan, 2015) does not mention AICDA as an important driver for B cell precursor ALL.

The reviewer is right that this excellent review by Hunger and Mullighan does not mention AICDA as a driver. The review is, however, more focussed on general aspects of B-ALL including risk stratification and treatment than biological pathogenesis. It was also published in the same year as the original work about AICDA in ALL by Swaminathan et. al. (Nat. Immunol. 16, 766–774, 2015) which may be another reason why it is not mentioned.

And the Immgen database (<http://www.immgen.org/databrowser/index.html>); search criteria “key Populations” and “aicda” shows that Aicda is not expressed in B cell precursors. This is supported by the authors figure 1, which also shows B cell precursors do not express Aicda. Taking all of these features into consideration, I do not think it is surprising that Aicda knockout has little effect on B cell precursor ALL. Post germinal center lymphoma are not equivalent to B cell precursor ALL.

We agree with the reviewer that B cell precursors do not constitutively express AICDA as shown by the Immgen database. However, AICDA expression has been reported to be induced in B cell precursors in response to several stimuli, e.g. LPS treatment (Swaminathan, S. et al. Nat. Immunol. 16, 766–774, 2015), cytokines (Greaves M. Nat Rev Cancer 18, 471-484, 2018) or virus infection (Gourzi P, et al., Immunity , 2006, 24:779-786). AID is frequently aberrantly expressed in cancer cells

(reviewed e.g. Pablo Pérez-Durán et al. *Carcinogenesis*, 28, 2427–2433, 2007, Table I, please see below), also in pre B ALL cells (Feldhahn N, et al. *J Exp Med.* 2007 14;204(5):1157-66; Gruber TA, et al. *Cancer Res.* 2010;70:7411-7420)

[REDACTED]

The analysis of WES data remains confusing. The authors compare matched germline and tumor, but find more variants in the germline than the tumor. Does this mean that the tumors have net negative acquired somatic changes? Or perhaps the filtering criteria are not adequately strict.

We thank the reviewer for calling our attention to this misleading presentation of our sequencing data in Figure 2a. Indeed for 3 of the 13 mice, we obtain slightly higher numbers of variants in the tumor sample (112 to 1045 variants, corresponding to 0.6 to 4.5 % of the variants called in the tumor sample) compared to the control sample in the first step of variant calling.

Mouse ID	Number of Variants (Tumor – Normal)	Corresponds to total number of tumor variants
I921	112	0.6%
V414	1045	4.5%
V517	225	1.25%

These variants are not somatic variants, but variants that are called when each sample is -as a first step in the bioinformatics pipeline- compared to the mouse reference genome GRCm38.71. Small differences in called variants are due to general aspects inherent to whole exome sequencing. The genomic DNA is fragmented by sonification generating random DNA fragments of about 250 bp. Exonic regions are then enriched by baits using the Agilent SureSelectXT Mouse All Exon Kit ensuring high coverage of all exonic regions included. Due to the random fragmentation of the DNA, the boundaries of the captured fragments differ and include e.g. also intronic regions or boundaries. Due to the washing procedure, there are also inevitably still residual amounts of genomic DNA left in all samples. These regions are sequenced with much lower coverage and differ between individual samples. Therefore in these regions variants may be called in a tumor sample but not in the corresponding matched normal control sample and vice versa.

To avoid a misinterpretation of our data we revised Figure 2a and removed these initial call numbers.

Reviewers' comments:

Reviewer #1 (Remarks to the Author):

The revised manuscript by Rodriguez-Hernandez has improved the WES section in figure 2a, but the statement in the text "We could identify between 14769 and 35381 single-nucleotide variations (SNVs) in the tumor samples of Pax5-het/Aid-het and Pax5-het/Aid-KO mice, as well as between 14792 and 39254 SNV in the paired germline control samples" remains misleading. The vast number of these variants are not true variants; they are either artefacts of the analysis or strain polymorphisms. The useful number is the number of acquired SNV shown in fig 2a. Why not report these in the text?

The finding that *Aicda* does not play a role in B cell precursor ALL is not surprising, and, in my opinion, does not reflect a significant advance. *Aicda* is not expressed in normal B cell precursors; therefore, one wouldn't expect the elimination of *Aicda* to effect the development of B cell precursor ALL. The authors view that *Aicda* was thought to be important for B cell precursor ALL seems to rest on the Swaminathan Nature Immunology 2015 paper; the remainder of the papers cited (22-25) are review articles, commentaries, and speculations. I looked at the Swaminathan paper, which showed *Aicda* was undetectable by Western blot in unmanipulated B cell precursors (Fig 2d). The authors of that paper detected low level *Aid* expression in B220+ BM cells by qRT-PCR, but the sorting strategy used did not use surface IgM negativity to exclude the possibility that the B220+ cells were mature B cells that had circulated back to the BM. The transplant experiments used 5 rounds of manipulation in vitro prior to transplant, and the authors didn't show evidence that the leukemia which arose was B cell precursor ALL (as opposed to a more mature B-ALL). Curiously, the B-ALL they did show did not seem to express the ETV6-RUNX1 fusion which was used to trigger the B-ALL (Supp Fig 8). Therefore, I don't think that the existing evidence that *Aicda* is important for B cell precursor ALL is particularly compelling.

Point-by-point response:

Referee #1:

The revised manuscript by Rodriguez-Hernandez has improved the WES section in figure 2a, but the statement in the text “We could identify between 14769 and 35381 single-nucleotide variations (SNVs) in the tumor samples of Pax5-het/Aid-het and Pax5-het/Aid-KO mice, as well as between 14792 and 39254 SNV in the paired germline control samples” remains misleading. The vast number of these variants are not true variants; they are either artefacts of the analysis or strain polymorphisms. The useful number is the number of acquired SNV shown in fig 2a. Why not report these in the text?

Following Referee’s advice, in the revised version we have reported the somatic SNVs identified in the tumor samples in the text (page 9).

The finding that Aicda does not play a role in B cell precursor ALL is not surprising, and, in my opinion, does not reflect a significant advance. Aicda is not expressed in normal B cell precursors; therefore, one wouldn’t expect the elimination of Aicda to effect the development of B cell precursor ALL. The authors view that Aicda was thought to be important for B cell precursor ALL seems to rest on the Swaminathan Nature Immunology 2015 paper; the remainder of the papers cited (22-25) are review articles, commentaries, and speculations. I looked at the Swaminathan paper, which showed Aicda was undetectable by Western blot in unmanipulated B cell precursors (Fig 2d). The authors of that paper detected low level Aid expression in B220+ BM cells by qRT-PCR, but the sorting strategy used did not use surface IgM negativity to exclude the possibility that the B220+ cells were mature B cells that had circulated back to the BM. The transplant experiments used 5 rounds of manipulation in vitro prior to transplant, and the authors didn’t show evidence that the leukemia which arose was B cell precursor ALL (as opposed to a more mature B-ALL). Curiously, the B-ALL they did show did not seem to express the ETV6-RUNX1 fusion which was used to trigger the B-ALL (Supp Fig 8). Therefore, I don’t think that the existing evidence that Aicda is important for B cell precursor ALL is particularly compelling.

We fully agree with the Referee and for this reason we think it is mandatory to unequivocally clarify if Aid does govern infection-driven B-ALL development, because it will have very important implications for potential prevention of the B-ALL in susceptible carriers. Aid expression levels were not upregulated in preleukemic precursor B cells (stage of B-cell development which drives the common, or B cell precursor, subtype of ALL) from mouse models prone to develop pB-ALL under

natural infection exposure. However, in the revised version we show that in vitro exposure of preleukemic Pax5-het, and Sca1-ETV6-RUNX1 precursor pro-B cells to different immune activation stimuli resulted in high levels of AID mRNA (**Extended Data Fig. 1b**). We think that our results unequivocally show that infectious stimuli can promote malignant B-cell leukemogenesis through AID-independent mechanisms.

REVIEWERS' COMMENTS:

Reviewer #1 (Remarks to the Author):

I have read the reviewers letter and revised manuscript, but the central criticism I previously raised remains.

I agree with the authors' conclusion—"We think that our results unequivocally show that infectious stimuli can promote malignant B-cell leukemogenesis through AID-independent mechanisms." However, I do not think that there was compelling evidence to support the hypothesis that B cell precursor ALL was dependent on AID in the first place. Consider the following:

- 1) The experimental evidence cited that AID was important came from a single paper that had potential flaws in the experimental design, as outlined in my prior review.
- 2) AID is not expressed, as any significant level, in B cell precursors. The experiment shown in the revised manuscript (ExFig 1A) confirms that AID is not detectable in B cell precursors isolated from WT or infected mice. The in vitro experiment (ExFig 1B) shows that AID mRNA is increased in isolated B cells treated under various conditions in vitro, however, protein levels are not shown, rather, relative RNA levels are shown. It is not clear that the increase in relative mRNA is physiologically relevant. For instance, an increase from a mean of 0.1 mRNA copies per cell to 1 copy per cell would be a 10-fold increase, but may not be physiologically relevant. Moreover, the isolation strategy cited (purified B220+ cells from BM) would include mature, sIgM+ B cells (which are well known to express AID) that had recirculated back to the BM; these sIgM+ B cells can be 2% or more of the B220+ BM cells.
- 3) If AID were important for B cell precursor ALL, one might expect to see the "fingerprints" of AID mutation in B cell precursor ALL cells, such as hypermutation or frequent translocations into the IG switch region. Patients with B cell precursor ALL have relatively few SNV, and rarely have chromosomal translocations into the IGH switch region.

Therefore, although I agree with the authors conclusion (stated above), I do not think that observation represents a significant advance in our understanding of B cell precursor ALL.

Point-by-point response:

Reviewer #1 (Remarks to the Author):

I have read the reviewers letter and revised manuscript, but the central criticism I previously raised remains.

I agree with the authors' conclusion—"We think that our results unequivocally show that infectious stimuli can promote malignant B-cell leukemogenesis through AID-independent mechanisms."

We greatly appreciate this kind appraisal of our study by Reviewer #1.

However, I do not think that there was compelling evidence to support the hypothesis that B cell precursor ALL was dependent on AID in the first place. Consider the following: 1) The experimental evidence cited that AID was important came from a single paper that had potential flaws in the experimental design, as outlined in my prior review.

However, the current model in the field of B-cell leukemogenesis holds that AID expression is induced in pre-leukemic B-cell precursor cells in response to infection and promotes in this case secondary genetic changes that may lead to subsequent leukemia development. This model is presented in a figure of a recent review in Nature Reviews of Cancer (Mel Greaves, 2018 Aug;18(8):471-484), where it is stated that "*More recently, my colleagues and I have screened a series of inflammatory cytokines for their ability to trigger AID expression in human B cell precursors. The most potent was TGF β (V. Cazzaniga, A. M. Ford and M. Greaves, unpublished observations)*".

2) AID is not expressed, as any significant level, in B cell precursors. The experiment shown in the revised manuscript (ExFig 1A) confirms that AID is not detectable in B cell precursors isolated from WT or infected mice. The in vitro experiment (ExFig 1B) shows that AID mRNA is increased in isolated B cells treated under various conditions in vitro, however, protein levels are not shown, rather, relative RNA levels are shown. It is not clear that the increase in relative mRNA is physiologically relevant. For instance, an increase from a mean of 0.1 mRNA copies per cell to 1 copy per cell would be a 10-fold increase, but may not be physiologically relevant. Moreover, the isolation strategy cited (purified B220+ cells from BM) would include mature, sIgM+ B cells (which are well known to express AID) that had recirculated back to the BM; these sIgM+ B cells can be 2% or more of the B220+ BM cells.

In page 16, under the method heading "*Real-time PCR quantification (Q-PCR) of Aicda*", we clearly indicate that expression of AID was analyzed in sorted BM precursor B (B220^{low} IgM⁻) cells of wild-type, Pax5-het and Sca1-ETV6-RUNX1 mice. Thus, the sorted B-cell precursor

population used is IgM negative. In the pro-B cell culture, we purified B220+ cells, but only IL-7 dependent pure pro-B cells are propagated in vitro (for details see Martin-Lorenzo et al. Cancer Discovery 2015). Thus, we measured Aid RNA expression in precursor B cells. Nevertheless, we agree with the reviewer that AID is not overexpressed in early bone marrow B cell precursors. However, given the clonal nature of leukemia, we cannot exclude that Aid would be overexpressed at single preleukemic precursor B cell. This point is further stressed in the Discussion of the revised manuscript, page 11.

3) If AID were important for B cell precursor ALL, one might expect to see the “fingerprints” of AID mutation in B cell precursor ALL cells, such as hypermutation or frequent translocations into the IG switch region. Patients with B cell precursor ALL have relatively few SNV, and rarely have chromosomal translocations into the IGH switch region.

We also agree with the reviewer in this point and for this reason we show in the manuscript that our findings were also substantiated in humans, since AID off-targets are not significantly enriched in human B-ALL drivers (**Supplementary Fig. 13, Supplementary Table VI**). This reviewer’s point is now included in the Discussion of the revised manuscript, page 12.

Therefore, although I agree with the authors conclusion (stated above), I do not think that observation represents a significant advance in our understanding of B cell precursor ALL.

We thank the reviewer for being agreed with our conclusions. However, we think that the discovery that infection-mediated native B-ALL takes place without AID contribution is really important to fix a new model in the field of B-cell leukemogenesis. This new model will help to understand why only a minority of healthy newborns harboring a preleukemic clone evolve to overt B-ALL and to develop approaches to prevent childhood BALL. In addition, these findings have also potential implications for the clinical use of PI3K δ or Bruton’s tyrosine kinase inhibitors in the treatment of B-ALL.